



**Global high resolution monthly pCO₂ climatology for the coastal ocean derived from**
**neural network interpolation**
Goulven G. Laruelle[1], Peter Landschützer[2], Nicolas Gruber[3], Jean-Louis Tison[1], Bruno
Delille[4], Pierre Regnier[1]
1. Department Geoscience, Environment & Society (DGES), Université Libre de
Bruxelles, Belgium
2. Max Planck Institute for Meteorology, Hamburg, Germany
3. Environmental Physics, Institute of Biogeochemistry and Pollutant Dynamics, ETH
Zürich, Switzerland
4. Unité d'Oceanographie Chimique, Astrophysics, Geophysics and Oceanography
department, University of Liège, Belgium
Corresponding author: Goulven G. Laruelle





**Abstract**
In spite of the recent strong increase in the number of measurements of the partial pressure of
$CO_2$ in the surface ocean (pCO$_2$), the air-sea $CO_2$ balance of the continental shelf seas remains
poorly quantified. This is a consequence of these regions remaining strongly under-sampled
both in time and space, and of surface pCO$_2$ exhibiting much higher temporal and spatial
variability in these regions compared to the open ocean. Here, we use a modified version of a
two-step artificial neural network method (SOM-FFN, Landschützer et al., 2013) to
interpolate the pCO$_2$ data along the continental margins with a spatial resolution of 0.25
degrees and with monthly resolution from 1998 until 2014. The most important modifications
compared to the original SOM-FFN method are (i) the much higher spatial resolution, and (ii)
the inclusion of sea-ice as a predictor of pCO$_2$. The validity of our interpolation, both in space
and time, is assessed by comparing the SOM-FFN outputs with pCO$_2$ measurements extracted
from the SOCATv3.0 and LDVEO2014 datasets. The new coastal pCO$_2$ product confirms a
previously suggested general meridional trend of the annual mean pCO$_2$ in all the continental
shelves with high values in the tropics and dropping to values beneath those of the atmosphere
at higher latitudes. But significant differences in the seasonality across the ocean basins exist.
The shelves of the western and northern Pacific, as well as the shelves in the temperate North
Atlantic display particularly pronounced seasonal variations in pCO$_2$, while the shelves in the
southeastern Atlantic and in the South Pacific reveal a much smaller seasonality. Overall, the
seasonality in shelf pCO$_2$ cannot solely be explained by temperature-induced changes in
solubility, but are also the result of seasonal changes in circulation, mixing, and biological
productivity. Finally, thanks to this product having been extended to cover open ocean areas



as well, it can be readily merged with existing global open ocean products to produce a true
global perspective of the spatial and temporal variability of surface ocean $pCO_2$.



**1. Introduction**
The quantitative contribution of the coastal ocean to the global oceanic uptake of atmospheric
$CO_2$ is still being debated (Borges et al., 2005; Chen and Borges, 2009; Cai, 2011;
Wanninkhof et al., 2013; Gruber, 2015), but several recent studies have suggested that the flux
density, or uptake per unit area, is greater over continental shelf seas than over the open ocean
(Chen et al., 2013; Laruelle et al., 2014). Laruelle et al. (2014) used more than $3 \cdot 10^6$ $pCO_2$
measurements from the SOCATv2 database (Pfeil et al., 2014 Bakker et al., 2016) to
demonstrate very strong disparities in air-seawater $CO_2$ exchange at the regional scale as well
as pronounced seasonal variations, especially at temperate latitudes. Furthermore, it was
suggested that despite the presence of a seasonally varying sea-ice cover, Arctic continental
shelves are a regional hotspot of $CO_2$ uptake (Bates et al., 2006; Laruelle et al., 2014;
Yasunaka et al., 2016). Yet, even with this much larger dataset compared to previous reports,
large regions of the global coastal ocean remained either devoid of data or very poorly
monitored in space and time, including the seasonal cycle. These data gaps do not only limit
our ability to reduce uncertainties in flux estimates and to unravel whether they differ from
the adjacent open ocean, but also hamper the identification and quantification of the many
underlying processes controlling the source-sink nature of the coastal ocean (Bauer et al.,
2013). Laruelle et al., (2014) attempted to overcome this limitation by combining various
upscaling methods depending on data density in different regions, e.g., resorted to using
annual means, wherever the seasonal coverage was deemed to be insufficient. But they could
not overcome the limitation that the data alone are insufficient to assess whether there are any
trends in coastal fluxes. This is a serious gap when considering that the influence of human
activity on coastal system is increasing rapidly (Doney, 2010; Cai, 2011; Regnier et al., 2013;





Gruber, 2015).
In the open ocean, novel statistical methods relying on artificial neural networks (ANNs) have
permitted the generation of a series of high-resolution continuous monthly maps for ocean
surface $CO_2$ partial pressures (p$CO_2$) (e.g., Landschützer et al., 2013; Sasse et al., 2013;
Nakaoka et al., 2013; Zeng et al., 2014). Although differing in their details (see e.g.,
Rödenbeck et al., 2015 for an overview), these products have typically a nominal spatial
resolution of 1-degree and monthly temporal resolution. By filling in the spatial and temporal
gaps, these products greatly facilitate the calculation of the air-sea $CO_2$ exchange, as they do
not require separate assumptions about the surface ocean p$CO_2$ in areas lacking data. Such
methods are also well suited to resolve spatial gradients, and they also permit to determine
seasonal and inter-annual variations and trends in p$CO_2$ (e.g., Landschützer et al., 2014, 2015,
2016; Zeng et al., 2014). Because of the small relative contribution of the coastal ocean to the
total oceanic surface area and the relatively coarse spatial resolution of the ANN-based
surface ocean p$CO_2$ products so far, they are not well suited to resolve the high
spatio-temporal variations of the surface ocean p$CO_2$ fields along the shelves.
Reproducing the complex seasonal dynamics of the $CO_2$ exchange at the air-water interface in
the coastal ocean is of particular importance considering that they often display large
intra-annual variability (Signorini et al., 2013). For instance, in temperate climates, it is
common for continental shelf waters to turn from $CO_2$ sinks for the atmosphere during spring
to $CO_2$ sources during summer (Shadwick et al., 2010; Cai, 2011; Laruelle et al., 2014, 2015).
Shelf waters are also typically characterized by small-scale physical features such as coastal
currents, river plumes and eddies inducing sharp biogeochemical fronts (Liu et al., 2010) that
markedly influence the spatial patterns of the p$CO_2$ fields (e.g., Turi et al., 2014).



To resolve the high spatial and temporal variability in air-sea $CO_2$ exchange over the global
shelf region, the two step artificial neural network method developed by Landschützer et al.
(2013) is modified here for the specific conditions that prevail in these environments. Our
calculations are performed at a much finer resolution of 0.25 degree and new environmental
drivers such as sea ice cover are used at high latitude to account for the potentially significant
role of sea-ice in the $CO_2$ exchange (Bates et al., 2006; Vancoppenolle et al., 2013; Parmentier
et al., 2013; Moreau et al., 2016; Grimm et al., 2016). The definition of the coastal/open
oceanic boundary significantly varies from one study to the other (Walsh, 1988; Laruelle et al.,
2013), with a potentially large impact on the shelf $CO_2$ budget (Laruelle et al., 2010). Here,
we use a very wide definition for this boundary (i.e., 300km width or 1000m depth) to secure
spatial continuity between our new shelf $pCO_2$ product and those already existing for the open
ocean (Landschützer et al., 2013, 2016; Rödenbeck et al., 2015). Our approach leads to the
first continuous and monthly resolved $pCO_2$ climatology (1998-2014) across the global shelf
region, permitting us to study the seasonal dynamics of these regions in relationship to that of
the adjacent open ocean.

**2. Methods**
The method used in this study is a modified version of the SOM-FFN method developed by
Landschützer et al. (2013) to calculate monthly-resolved $pCO_2$ maps of the Atlantic Ocean at
a 1 degree resolution and later applied to the entire global open ocean (Landschützer et al.,
2014). The reconstruction of a continuous $pCO_2$ field involves establishing numerical
relationships between $pCO_2$ and a number of independent environmental predictors that are
known to control its variability both in time and space. The first step of the method relies on



the use of a neural network clustering algorithm (Self Organizing Map, SOM) to define a
discrete set of biogeochemical provinces characterized by similar relationships between the
independent environmental variables and a climatological $pCO_2$ field. The second step
consists in deriving non-linear and continuous relationships between $pCO_2$ and some or all of
the aforementioned independent variables using a feed-forward network (FFN) method,
within each biogeochemical province created by the SOM. The method is extensively
described in Landschützer et al. (2013, 2015) but the specific modifications introduced in this
study to better simulate the characteristics of the shelves, the choice of environmental drivers
and their data sources as well as the definition of the geographic extent of this analysis are
described in the following sections.

**2.1. Modifications of the SOM-FFN method**
The specific characteristics of the continental shelves motivated a number of modifications of
the global ocean SOM-FFN method, including a 16 fold increase in spatial resolution from 1
degree to 0.25 degree, the introduction of a second neuron layer in the FFN calculations, the
addition of new environmental variables as biogeochemical predictors, and a shortening of the
simulation period to the period 1998 through 2014. All these modifications are detailed here
below.

The higher resolution of 0.25°×0.25° results in over 2 million grid cells that help in better

tracking the global coastline and its complex geomorphological features (Crossland et al.,
2005; Liu, 2010). It is also common along Eastern and Western boundary currents to find
continental shelves as narrow as 10-20 km, an extension that is thus significantly smaller than
a single cell at 1-degree resolution. Additionally, biogeochemical fronts associated to river



plumes, coastal currents and upwelling are characterized by spatial scales of the order of tens
of kilometers or even smaller (Wijesekera et al., 2003). The chosen resolution is also identical
to the gridded coastal $pCO_2$ product from the SOCAT initiative (Sabine et al, 2013, Bakker et
al., 2014).

The definition of the geographic extent of the shelf region excludes estuaries and other

inland water bodies, but uses a wide limit for the outer continental shelf that encapsulates all
current definitions of the coastal ocean. This approach facilitates future integration with
existing global ocean data products (e.g., Landschützer et al., 2016; Rödenbeck et al., 2015)
and model outputs, which typically struggle to represent the shallowest parts of the ocean
(Bourgeois et al., 2016). The outer limit used here is given by whichever point is the furthest
from the coast: either 300km distance from the coastline (which roughly corresponds to the
outer edge of territorial waters (Crossland et al., 2005)) or the 1000m isobaths (Laruelle et al.,
2013). The resulting domain (Fig SI1) covers 77 million $km^2$, more than twice the surface
area generally attributed to the coastal ocean (Walsh et al., 1998; Liu et al., 2010; Laruelle et
al., 2013).

The predictor variables for the SOM-FFN networks were chosen based on a set of trial

and error experiments with the selection criteria being the quality of fit, i.e., the best
reconstruction of the available observations. The first step of the SOM-FFN calculations, i.e.,
the self-organizing map-based clustering (SOM) relies on the assignment of the surface ocean
data to biogeochemical provinces sharing common spatio-temporal patterns of sea-surface
temperature (SST), sea-surface salinity (SSS), bathymetry, rate of change in sea ice coverage
and observed $pCO_2$. The use of the rate of change in monthly sea ice concentration is a
novelty compared to the set-up of Landschützer et al. (2013) and is calculated from the





gridded monthly sea ice concentration field of Cavalieri et al. (1996). It allows accounting for
the complex processes occurring in melting and forming sea ice that are known to strongly
influence the dynamics of the carbon within sea-ice covered areas (Parmentier et al., 2013).
This first step is performed without any data normalization of the datasets. Based on a series
of simulations using different numbers of biogeochemical provinces, we found that a
clustering of the data into 10 biogeochemical provinces minimized the average deviation
between simulated and observed $pCO_2$ (see below).

During the second step of the calculation, i.e., the application of the feed-forward

network method (FFN), SST, SSS, bathymetry, sea-ice concentration and chlorophyll a are
used as predictors to establish the non-linear relationship between these predictors and the
target $pCO_2$ (for data sources, see below). Similar to the SOM in step one, the selected
variables not only comprise proxies representing the solubility and biological pumps of the
coastal ocean, but also yield the best fit to the data. These calculations are done iteratively on
an incomplete dataset in order to perform an assessment on the remaining data after each
iteration, until an optimal relationship is found. This step now includes a second artificial
neuron layer that consists, for each iteration, of an additional procedure of optimization of the
relationship fitting. This addition significantly increases the calculation time but prevents the
SOM-FFN from generating negative values. Additionally, as performed in Landschützer et al.
(2015), the output $pCO_2$ data were smoothed using the spatial and temporal mean of each
point's neighboring pixels both in time and space within the 3 pixel neighborhood domain.
This operation is performed iteratively and does not significantly alter the results, but it
ensures smoother transitions in the $pCO_2$ field at the boundaries between the provinces. This
smoothing method yielded good results for the open Southern Ocean where marked $pCO_2$



fronts are also observed (Landschützer et al., 2015) and was deemed relevant here due to the
potentially strong $pCO_2$ gradients characterizing the shelves.

Another change from the most recent global ocean SOM-FFN application (Landschützer

et al., 2016) is the different temporal extension of the simulation period, which covers the
period from 1998 through 2014 only, instead of 1982 through 2011. This overall shortening
was necessary because one of environmental driver, the chlorophyll data derived from
SeaWIFS, only starts in September 1997 (NASA, 2016). Monthly chlorophyll data throughout
the entire simulation period was preferred here over the use of a monthly climatology as done
in Landschützer et al. (2016) to better capture inter-annual variability.

**2.2. Data Sources and processing**
All the datasets used in our calculations were converted from their original spatial resolutions
to a regular 0.25 degree resolution grid. The temporal resolution of all datasets is monthly (i.e.,
204 months over the entire period), except for the bathymetry that is assumed constant over
the course of the simulations. SST and SSS maps were taken from the World Ocean Atlas
(Antonov et al., 2010 for SST and Locarnini et al., 2010 for SSS). The bathymetry was
extracted from the global ETOPO2 database (US Department of Commerce, 2006). The sea
ice concentrations are recalculated from the global 25 km resolution monthly data product
compiled by the NSIDC (National Snow and Ice Cover Data; Cavalieri et al., 1996). The
chlorophyll surface concentrations were extracted from the monthly 9 km resolution SeaWIFS
data product (NASA, 2016). Finally, the surface ocean $pCO_2$ were taken from the gridded
SOCATv3 product (Sabine et al., 2013; Bakker et al., 2016) while those used from the
validation are extracted from the LDEOv2014 database (Takahashi et al., 2016). This latter





database contains ~ 10.5 million $pCO_2$ measurements collected between 1957 and 2015.
While a large overlap with the SOCAT database is inevitable, LDEOv2014 was compiled
independently and is the only other global $pCO_2$ dataset (of comparable size and coverage to
SOCAT) presently available. The data from SOCAT were converted from $fCO_2$ (fugacity of
$CO_2$ in water) into $pCO_2$ using the formulation reported in Takahashi et al. (2012).

**2.3. Evaluation procedures**
We evaluated the coastal SOM-FFN product using the root mean squared error (RMSE)
metric, calculated as the difference between estimated and observed $pCO_2$. During the
development stage, preliminary simulations were performed using only data from SOCAT
v2.0 (Pfeil et al., 2013, Sabine et al. 2013) to train the FFN algorithm. Each simulation was
carried out using different subsets of environmental predictors extracted from the complete set
(SST, SSS, bathymetry, sea ice concentration and chlorophyll a). The results obtained were
then compared to the more complete dataset of SOCAT v3.0, which contain 40% more shelf
$pCO_2$ measurements from 1998 through 2014 (Bakker et al., 2016). This process allowed, for
each province, to calculate the RMSE for several combinations of independent predictor
variables for the $pCO_2$. Next, the combinations of predictors displaying the lowest RMSE
were kept for the final simulations, which then used all data from SOCAT v3.0. Thus, the
$pCO_2$ calculations in each province potentially rely on a different set of predictors (Table 1).
The coastal SOM-FFN results are validated through a comparison with the LDEOv2014 data
base (Takahashi et al., 2016). Additionally, a model-to-model comparison is also performed
with the global ocean results of Landschützer et al. (2016) in the regions where the domains
overlap. To perform this latter analysis, the coastal high resolution coastal $pCO_2$ product



generated here was aggregated to a regular monthly 1° resolution to match the grid used by
Landschützer et al. (2016).
Finally, the ability of the coastal SOM-FFN to capture seasonal variations is assessed by
comparing the cell-average simulated monthly $pCO_2$ to monthly means for cells extracted
from the LDEOv2014 database. The cells retained for this analysis are all those for which the
average for each month could be calculated from measurements performed on at least three
different years.

**3. Results and discussion**
**3.1. Biogeochemical provinces**
Despite the fact that the SOM is not given any prior knowledge regarding space and time,
the spatial distribution of the 10 biogeochemical provinces is mostly controlled by latitudinal
gradients and distance from the coast (Figure 1; high-resolution monthly maps are also
available in the supplementary information (SI)). Although the exact spatial extent of each
province varies from one month to the other following the seasonal variations of the
environmental forcing parameters, each province roughly corresponds to one type of
climatological setting. Nevertheless, because of these spatial migrations, most cells belong to
different provinces depending on the month (see figure 1 of SI). These seasonal migrations
are mostly driven by changes in temperature, sea-ice cover and, to a lesser degree, salinity. P1
(Province 1, etc.) and P2 are the two largest provinces, covering $26.1 \cdot 10^6$ km$^2$ and
representing warm tropical regions with bottoms at shallow to intermediate depths. During
summer, the spatial coverage of P1 expands north- and southward as a consequence of
warming. P3 and P4 represent tropical regions with deeper bottom depths. They display less





seasonal changes in their spatial distribution than P1 due to weaker seasonal temperature
changes. P5 and P6 cover a combined $14 \cdot 10^6$ km$^2$ and correspond to sub-polar and temperate
regions, respectively. Their spatial distributions are subject to marked latitudinal migrations
throughout the year as a result of the large amplitude changes in seasonal temperature
observed in mid-latitude coastal waters (Laruelle et al., 2014). P7, P8, P9 and P10 together
cover $21.3 \cdot 10^6$ km$^2$. These provinces are partly (seasonally) covered by sea-ice with an
average ice cover of 41% and 65% for P7 and P10, respectively. P7 includes large fractions of
the enclosed seas at higher northern latitudes such as the Baltic Sea and Hudson Bay while
P10 (only $2.9 \cdot 10^6$ km$^2$) represents permanently deep and cold polar regions. P8 and P9
represent most of the polar shelves (both the Arctic and Antarctic) and are covered in sea ice
at levels of 47% and 56%, respectively. The regions experiencing most notable shifts in
province allocation during the year include the northern polar regions as well as the temperate
narrow shelves of the Atlantic and Pacific, particularly Western Europe and Eastern North
America and Eastern Asia (see Fig. SI1).

**3.2. Performance of the coastal SOM-FFN**
The mean climatological pCO$_2$ estimated by the coastal SOM-FFN for annually and
seasonally averaged conditions are reported in Figure 2. Before briefly analysing the main
spatial and temporal variability of the pCO$_2$ fields (section 3.3), we evaluate here the overall
performance of our interpolation method globally and at the level of each province, including
its ability to capture the seasonal cycle.
**3.2.1. Comparison with training SOCAT v3.0 data**




Within each province, the pCO$_2$ simulated by the coastal SOM-FFN are compared to the
measurements extracted from SOCAT v3.0 (table 2). Globally, the average difference between
observed and simulated pCO$_2$ is almost null (overall bias = +0.1 µatm). The average RMSE
over all provinces of 32.6 µatm is comparable with those reported for other statistical
reconstruction of coastal pCO$_2$ fields although none of these studies were performed at global
scale (Chen et al., 2016). This RMSE is about twice that achieved for the open ocean
(Landschützer et al., 2014) reflecting the larger spatiotemporal variability in the coastal ocean,
as well as more complex processes governing that variability. Considering these complexities,
the achieved RMSE is quite good.
Significant variations in both bias and RMSE can be observed between provinces (table 2). P2
and P3 have the best fit between simulated and observed pCO$_2$ with absolute bias and RMSE
lower than 2 µatm and 20 µatm, respectively. In 6 provinces which cover a cumulated surface
area of 52.6 10$^6$ km$^2$ (P1, P2, P3, P4, P6 and P8) RMSE's do not exceed 30 µatm. In P7
however, bias and RMSE are maximum with values of 7.4 µatm and 63.4 µatm, respectively.
Overall, the performance of the SOM-FFN deteriorates for provinces regularly covered by
sea-ice ice (P7-10) and in which data coverage is relatively low.
**3.2.2. Comparison with LDEOv2014 data**
The comparison of our results with the data from LDEOv2014 yields a very small bias of -2.4
µatm (calculated as the average difference between observed and SOM-FFN estimated pCO$_2$)
for the entire shelf domain. However, the spread is relatively large with an average RMSE of
42 µatm. This average RMSE is 24% larger than the one obtained when comparing the
SOM-FFN results with the SOCAT dataset, which has been used to train the model. A
province-based analysis reveals strong differences in the calculated RMSEs, ranging from 20



µatm to 67 µatm (Table 2, LDEO). A review of various statistical models used to generate
continuous global ocean pCO$_2$ maps reports RMSE or uncertainties typically varying within
the 10-35 µatm range (Chen et al., 2016) with outliers as high as 50 µatm in the Mississippi
delta (Lohrenz and Cai, 2006). This report shows that open ocean estimates generally yields
RMSE lower than 17 µatm, in agreement with Landschützer et al. (2014), whereas coastal
estimates are associated with much higher uncertainties. This is likely because these coastal
regions have complex biogeochemical dynamics and high frequency variability that cannot be
fully captured with the current generation of data interpolation techniques using the limited
available predictor data.
In our simulations, provinces P1, P2 and P4 have negligible biases (with absolute values <0.5
µatm, table 2) and RMSE < 30 µatm, which compares with the most robust pCO$_2$ regional
coastal estimates from the literature (Chen et al., 2016). Together, these 3 provinces account
for 44% of our domain. P3 and P5 display slightly higher biases of -2.3 and -5.2 µatm,
respectively and RMSE of 44 and 67 µatm. Overall, these 5 provinces covering the tropical
and temperate latitudinal bands account for >62% of the shelf surface area and yield RMSE of
less than 45 µatm and absolute biases of less than 4 µatm. Provinces in the sub-polar and
polar regions (P6, P7, P8, P9 and P10) overall display larger deviations with respect to the
LDEOv2014 dataset, but the absolute value of their biases never exceeds 10 µatm. Except for
P8, which displays a RMSE of 35 µatm, all other provinces are characterized by RMSE
falling in the 45-70 µatm range. This suggests a significantly lower performance of the
SOM-FFN in regions partly covered in sea-ice. This can be attributed to the limited number of
available data points and their very heterogeneous distribution in time and space, as well as to
the very limited range of variation of some of the controlling variable such as temperature and



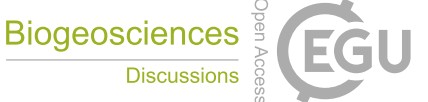

salinity. The relatively good performance of the model in tropical region might be partly
attributed to the relatively small seasonal variations in $pCO_2$ within these areas.

While the use of RMSE provides a valid quantitative assessment of the model

performance, it does not provide insights regarding its ability to reproduce the seasonal $pCO_2$
cycle. To address this issue, Figure 3 displays observed mean monthly $pCO_2$ extracted from
LDEOv2014 and calculated by the coastal SOM-FFN for the 53 locations where the
LDEOv2014 database has the most data (>40 month). The error bars associated with the
observations reflect the inter-annual variability. Overall, the coastal SOM-FFN captures the
timing of the seasonal $pCO_2$ cycle in most locations well with $pCO_2$ minima and maxima
occurring at the same time in our results and in the LDEOv2014 data.
The $pCO_2$ maximum generally taking place in early summer is the most accurately captured
by the coastal SOM-FFN. In terms of amplitudes in the $pCO_2$ signal, the coastal SOM-FFN
and the LDEOv2014 data reveal how different the seasonal $pCO_2$ cycle is from one region to
the other, with very low amplitude (<40 µatm) in some sub-tropical areas, amplitudes > 100
µatm at high Northern and Southern latitudes, and sometimes very sharp increases during
summer like off the coast of Japan. In most regions, the SOM-FFN-based reconstructions are
able to capture these variations and predict seasonal amplitudes comparable to that observed
in the data. However, in cells for which the difference between observed and simulated
seasonal $pCO_2$ amplitude is larger than 20%, the coastal SOM-FFN tends to systematically
underestimate the amplitude of the seasonal $pCO_2$ cycle. This limitation of our model might
result from the often short time scales associated with the continental influences in near-shore
locations, which are not captured by the environmental predictors used in our calculation. It

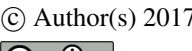



may also be the results of very short-term events that are aliased in our monthly average
calculations.

### 3.2.3. Comparison with global SOM-FFN

The comparison of our coastal SOM-FFN results with those of Landschützer et al. (2016) for
the overlapping grid cells (Table 2) reveals significant differences between both interpolated
data products with a RMSE between 20 and 37 µatm for most provinces except P7 and P9 (53
and 55 µatm, respectively). These RMSE values are comparable, but slightly lower than those
obtained for the comparison with the LDEO_v2014 database. The differences (coastal
SOM-FFN minus global SOM-FFN), however, are larger than those observed between our
results and the LDEO_v2014 database and highlight the current knowledge gap regarding the
mean state and variability of the transition zone. They range from -17.6 to 8.6 µatm from one
province to the other but only amount to -0.6 µatm when considering the cells from all
provinces.
The overlaping cells used for the comparison with Landschützer et al. (2016) are mostly
located over 100km away from the coastline and therefore the open ocean as well as our new
shelf ocean data set are constrained by fairly different data because all the 'shelf' cells from
the open ocean data product have a $pCO_2$ calculated by a model calibrated mostly for
conditions representative of the open ocean. Our results indicate that the very nearshore
processes controlling the $CO_2$ dynamics likely are the most difficult to reproduce with a
global SOM-FFN.

### 3.3. Spatial and temporal variability of the coastal $pCO_2$

### 3.3.1 Spatial variability



Figure 2a presents the annual average $pCO_2$ estimated by the coastal SOM-FFN, representing
the mean over 1998 through 2014 period (monthly climatological maps are shown in Fig. SI
2). High annual mean values of $pCO_2$, close to or above atmospheric levels, are estimated
around the equator up to the tropics. This is consistent with previous studies that identified
tropical and equatorial coastal regions as weak $CO_2$ sources for the atmosphere (Borges et al.,
2005; Cai, 2011; Laruelle et al., 2010; 2014). A hotspot of very high $pCO_2$ emerges from our
analysis in the Indian Ocean, extending past the tropic of Cancer into the eastern
Mediterranean Sea as well as the Red Sea and the Persian Gulf. These regions are poorly
monitored and it remains difficult to assess if $pCO_2$ values in excess of 450 µatm are realistic
or not, but the limited body of available literature suggests that very high $pCO_2$ are indeed
observed in these regions (Ali, 2008; Omer, 2010). The very high temperature and salinity
conditions observed in the Red Sea, in particular, reduce the $CO_2$ solubility and induce very
high $pCO_2$ conditions. However, these predicted $pCO_2$ lie outside of the range used for the
calibration of the SOM-FFN (typically 200-450 µatm) and should thus be considered with
caution.
In both hemispheres, $pCO_2$ in the 320 to 360 µatm range are generally reconstructed at
temperate latitudes, i.e., up to 50°N and 50°S, respectively. The northern high latitudes
generally have very low $pCO_2$ values, down to 300 µatm and below, a result that is consistent
with the Arctic shelves contributing a large proportion (up to 60%) of the global coastal
carbon sink (Bates and Mathis, 2009; Cai, 2011; Laruelle et al., 2014). Several hotspots of
$pCO_2$ with values as high as 450 µatm can be observed nevertheless north of 70°N, most
notably along the eastern coast of Siberia in winter (see Fig. SI 3), which displays a large zone
characterized by $pCO_2 > 400$ µatm centred around the mouth of the Kolyma River. Such high



$pCO_2$ values have been punctually observed in Arctic coastal waters (Anderson et al., 2009)
and could result from the discharge of highly oversaturated riverine waters. But, overall,
$pCO_2$ measurements over Siberian shelves are particularly rare. Thus, our results should be
considered with caution in this region because of the scarcity of data to train and validate the
coastal SOM-FFN. It should also be noted that the vast majority of this high $pCO_2$ region is
covered by sea ice (Fig. 2b&c) and, although the model estimates $pCO_2$ values over the entire
domain, only ice-free (or partially ice-free) cells will contribute to the $CO_2$ exchange across
the air-sea interface (Bates and Mathis, 2009; Laruelle et al., 2014).
**3.3.2. Temporal variability**
The reconstructed $pCO_2$ field is also subject to large seasonal variations (see figures SI 2&3).
To explore these variations further, Figure 4 reports seasonal-mean latitudinal profiles of
$pCO_2$ for continental shelves neighbouring the Eastern Pacific, Atlantic, Indian and Western
Pacific, respectively. The analysis excludes continental shelves at latitudes higher than 65
degrees because a large fraction of these shelves are seasonally covered by sea ice. The
latitudinal $pCO_2$ profiles reveal that, in most regions, highest and lowest $pCO_2$ values are
observed during the warmest and coldest months, respectively. This trend is particularly
pronounced at temperate latitudes where the seasonal $pCO_2$ amplitude can reach 60μatm and
is exemplified by regions such as the western Mediterranean Sea or the eastern coast of
America, which become supersaturated in $CO_2$ compared to the atmosphere during the
summer months. There are, however, a few other regions, where the lowest $pCO_2$ is found in
the summer, such as the Baltic Sea (Thomas and Schneider, 1999). Around the equator, the
magnitude of the seasonal variations in $pCO_2$ is limited and does not exceed 30 μatm.

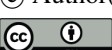



Although the general latitudinal trend of the annual mean $pCO_2$ is similar in all the continental
shelves, significant differences in the seasonality can be observed across the largest ocean
basins. In particular, most of the East Pacific shelves display limited seasonal change in $pCO_2$
(typically below 40 µatm) while the West Pacific shelves have seasonal $pCO_2$ amplitudes that
can exceed 60 µatm in temperate regions and 100 µatm at high latitudes (above 55° N). Along
the Atlantic shelves, the seasonal signal is more pronounced in the north compared to the
south, in agreement with Laruelle et al. (2014). Overall, the North Pacific (north of 55°N)
displays the most pronounced seasonal change in $pCO_2$ with a difference of 80 µatm between
summer and winter. In the Indian Ocean, the seasonal dynamics of $pCO_2$ is partly regulated by
seasonal upwelling induced by the Monsoon (Liu et al., 2010). In this basin above the equator,
April, May and June are the month displaying the highest $pCO_2$ and the seasonal variations do
not exceed 40 µatm. In contrast, the seasonal cycle is quite pronounced in the Indian Ocean
south of the equator (~60 µatm).

Latitudinal profiles of SST (Fig 4, bottom) are similar in all coastal oceans and display

minimal seasonal variations around the equator and amplitudes as large as 20°C at temperate
latitudes. The comparison between $pCO_2$ and SST profiles allows us to assess the contribution
of temperature-induced changes in $CO_2$ solubility to the seasonal $pCO_2$ variability in the
continental shelf waters. However, other factors such as seasonal upwelling and biological
activity also strongly influence coastal $pCO_2$ and contribute to the complexity of the seasonal
$pCO_2$ profiles. To quantify the effect of temperature on seasonal variations of $pCO_2$, the latter
is normalized to the mean temperature at different latitudes in each oceanic basin (Fig. 5)
using the formula proposed by Takahashi et al. (1993):

$$npCO_2 = pCO_{2,obs} \times exp\big(0.0423 \times (T_{mean} - T_{obs})\big) \qquad (1)$$





where $npCO_2$ is the temperature normalized pCO$_2$, $pCO_{2,obs}$ is the observed pCO$_2$ at the
observed temperature T$_{obs}$ and T$_{mean}$ is the yearly mean temperature at the considered location.
In sea-water, an increase in water temperature induces a decrease in gas solubility which leads
to a higher water pCO$_2$. Thus, comparing npCO$_2$ with observed pCO$_2$ monthly values
provides a quantitative estimate of the influence of seasonal temperature change on the
seasonality of pCO$_2$.
For all latitudes and oceanic basins, pCO$_2$ is minimum in late winter or early spring, i.e., at
the time when npCO$_2$ has its maximum. pCO$_2$ also generally displays a maximum in summer,
while npCO$_2$ reaches its minimum then (Fig. 5). The amplitude of the changes in npCO$_2$ is
quite consistent across oceans and about 2 to 3 times larger than that of pCO$_2$. Between 45°N
and 60° N, the variations in npCO$_2$ largely exceed 100 µatm (up to 180 µatm at 60° N in the
West Pacific). In these regions, the magnitude of the seasonal temperature changes reaches 20°
C between winter and summer (Fig. 4). A seasonal signal in pCO$_2$ with a minimum in late
winter or spring when npCO$_2$ is maximal can also be identified. However, the magnitude of
the seasonal variations in pCO$_2$ is significantly smaller than those of npCO$_2$, suggesting that
other processes such as biological uptake or transport partly offsets the temperature effect on
solubility. In the subpolar western Pacific shelves (60° N), a second pronounced dip in pCO$_2$
following a weaker one in spring is observed in summer, which suggests the occurrence of a
pronounced summer biological activity taking up large amounts of CO$_2$. This would also
explain the sharp increase in pCO$_2$ in the following month, as a result of the degradation of
organic matter synthesized during the summer bloom. This region is also the one subjected to
the strongest seasonal temperature gradient as evidenced by the amplitude of the seasonal
npCO$_2$ which reaches 200µatm. At 20° N, the amplitude of the changes in both pCO$_2$ and



$npCO_2$ are lower than at higher latitudes. $pCO_2$ varies by ~30µatm between summer and
winter in all oceanic basin while the seasonal variations in $npCO_2$ are more pronounced in the
Pacific (~60µatm) than in the Atlantic or the Indian Oceans. In the Southern Hemisphere, the
seasonal variations in $pCO_2$ are not as pronounced as in the Northern Hemisphere suggesting
that the changes induced by the solubility pump are compensated by biological activities. At
10°S and 30° S, the seasonal variations in $pCO_2$ rarely exceed 30 µatm in either basin with a
minimum observed around August.

**4. Summary**
This study presents the first global high-resolution monthly $pCO_2$ maps for continental shelf
waters at an unprecedented 0.25° spatial resolution. We show that when tailored for the
specific conditions of shelf systems, the SOM-FFN method previously employed in the open
ocean is capable of reproducing well-known and well-observed features of the $pCO_2$ field in
the coastal ocean. Our continuous, shelf product allows, for the first time, to analyze the
dominant spatial patterns of $pCO_2$ across all ocean basins and their seasonality. The data
product associated to this manuscript consists of a netcdf file containing the $pCO_2$ for ice-free
cells at a 0.25° spatial resolution for each of the 204 month of the simulation period (from
January 1998 to December 2014). Climatologically averaged $pCO_2$ maps for each month are
also provided. This data product can be combined with wind field products such as
ERA-interim (Dee, 2010) or CCMP (Atlas et al., 2011) to compute spatially and temporally
resolved air-sea $CO_2$ fluxes across the global shelf region, including the Arctic. Maps
including $pCO_2$ for ice covered cells are also available but should be treated with care because
the dynamics of $CO_2$ fluxes through sea ice are still poorly understood and air-sea gas transfer



velocities in partially sea ice covered areas cannot be predicted from classical wind speed
relationships (Lovely et al. 2015)

**5. Acknowledgements**
G. G. Laruelle and B Delille are postdoctoral researcher and research associate, respectively,
of F.R.S.-FNRS. The Surface Ocean $CO_2$ Atlas (SOCAT) is an international effort, supported
by the International Ocean Carbon Coordination Project (IOCCP), the Surface Ocean Lower
Atmosphere Study (SOLAS), and the Integrated Marine Biogeochemistry and Ecosystem
Research program (IMBER), in order to deliver a uniformly quality-controlled surface ocean
$CO_2$ database. The many researchers and funding agencies responsible for the collection of
data and quality control are thanked for their contributions to SOCAT. The research leading to
these results has received funding from the European Union's Horizon 2020 research and
innovation program under the Marie Sklodowska-Curie grant agreement No 643052 744
(C-CASCADES project). NG acknowledges support by ETH Zürich.





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





Table 1: List of the biogeochemical provinces, their geographic distribution and the
environmental predictors used to calculate surface ocean pCO$_2$. SSS stands for sea surface
salinity, SST for sea surface temperature, Bathy for bathymetry, Ice for sea-ice cover and Chl
for chlorophyll concentration.

| Province | Distribution | SSS | SST | Bathy | Ice | Chl |
|----------|--------------|-----|-----|-------|-----|-----|
| **P1** | Shallow tropical | X | X | X | | |
| **P2** | Tropical | X | X | X | X | X |
| **P3** | Deep Tropical | X | X | X | X | X |
| **P4** | Deep Tropical | X | X | X | X | X |
| **P5** | Sub Polar | X | X | X | X | |
| **P6** | Deep Temperate | X | X | X | X | X |
| **P7** | Shallow Polar | X | X | X | X | |
| **P8** | Deep Polar | X | X | X | X | |
| **P9** | Polar | X | X | X | | |
| **P10** | Very deep Polar | X | X | X | X | |






Table 2: Root mean squared error between observed and calculated pCO$_2$ in the different biogeochemical provinces. The SOM-FFN results are compared to data extracted from the LDEO database (Takahashi et al, 2014) and the overlapping cells from the Landschützer et al. (2016) pCO$_2$ climatology.

| Province | Surface Area (km$^2$) | Ice Cover (%) | SOCAT v3.0 Bias (µatm) | RMSE (µatm) | Landschützer 2016 Bias (µatm) | RMSE (µatm) | LDEO Bias (µatm) | RMSE (µatm) |
|---|---|---|---|---|---|---|---|---|
| P1 | 15.5 10$^6$ | 0 | -5.4 | 27.8 | 8.6 | 26.5 | -0.4 | 29.3 |
| P2 | 10.6 10$^6$ | 0 | 1.6 | 17.9 | 5.4 | 24.5 | -0.2 | 24.1 |
| P3 | 7.4 10$^6$ | 0 | 1.8 | 25.7 | 3.1 | 23.6 | -2.3 | 43.7 |
| P4 | 8.1 10$^6$ | 0 | -0.8 | 15.4 | 0.2 | 28.5 | -0.3 | 20.4 |
| P5 | 7.8 10$^6$ | 0.2 | -2.1 | 41.0 | -3.7 | 32.3 | -5.2 | 66.6 |
| P6 | 6.2 10$^6$ | 0 | -1.1 | 29.8 | -10.2 | 30.2 | -3.4 | 33.3 |
| P7 | 3.7 10$^6$ | 41.3 | 7.6 | 63.4 | -11.7 | 53.1 | -6.9 | 66.1 |
| P8 | 4.9 10$^6$ | 47.9 | -1.8 | 30.0 | 2.9 | 27.2 | -5.3 | 34.9 |
| P9 | 9.8 10$^6$ | 56.4 | 0.1 | 36.4 | -17.9 | 55.4 | -9.5 | 49.5 |
| P10 | 2.9 10$^6$ | 64.6 | 1.3 | 38.3 | 6.7 | 37.0 | -9.9 | 48.3 |
| | 76.9 10$^6$ | | 0.1 | 32.6 | -0.6 | 32.7 | -2.4 | 41.6 |





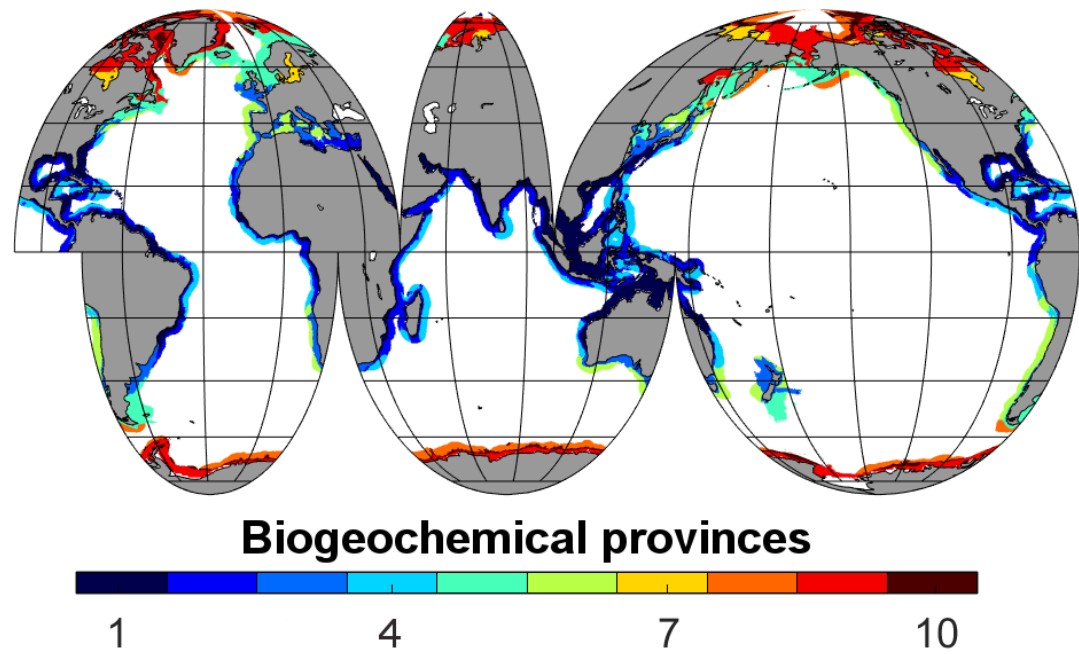

Figure 1: Map of the 10 different biogeochemical provinces generated by the artificial neural network method SOM-FFN.





Figure 2: Climatological mean $pCO_2$ for (a) the long-term averaged $pCO_2$ (rainbow color scale) and sea-ice coverage (black-white color scale). The long-term average $pCO_2$ corresponds to roughly the nominal year 2006, as the average was formed over the full analysis period from 1998 through 2014; (b) the months of January, February and March; and (c) the months of July, August and September.



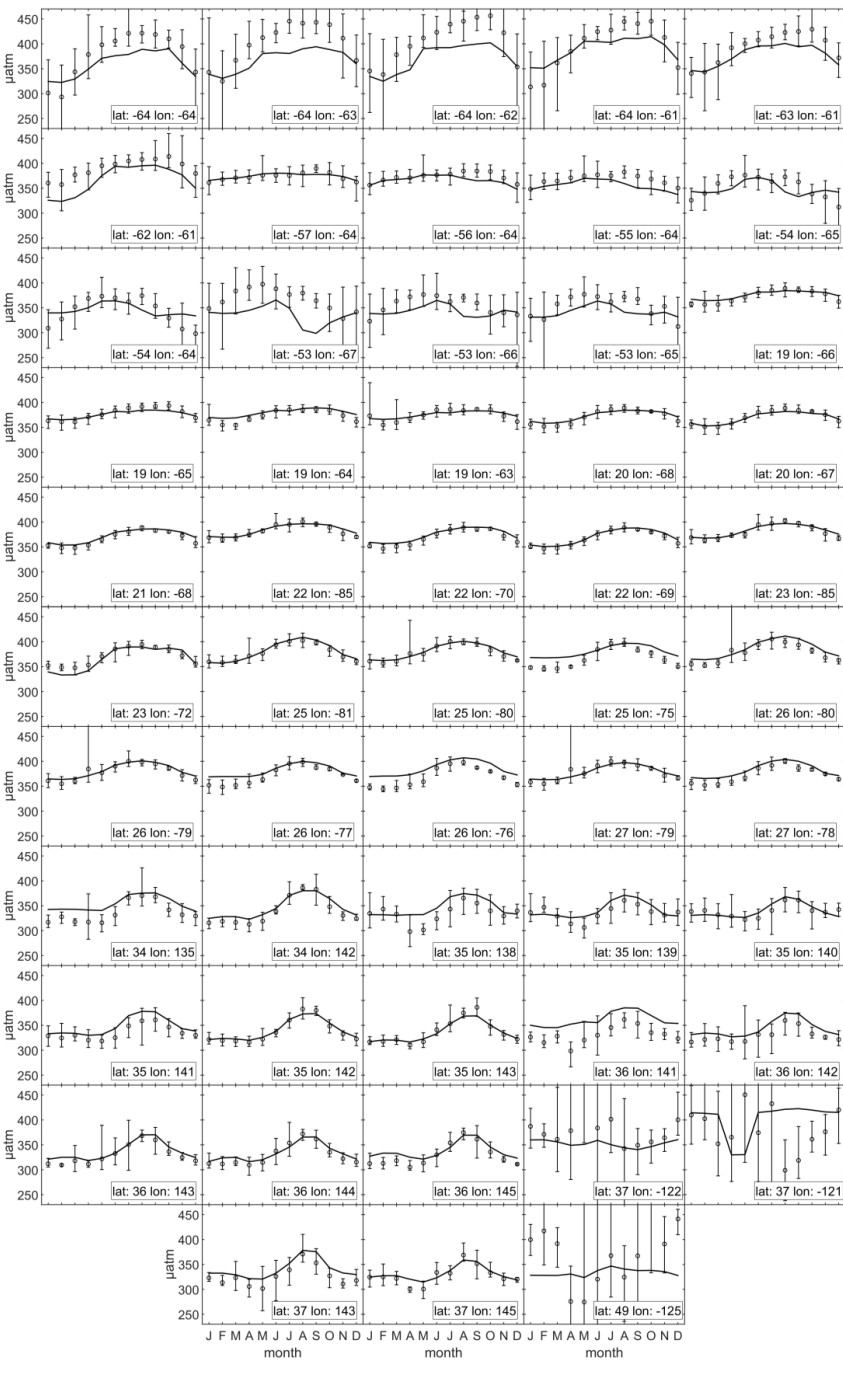

Figure 3: Climatological monthly mean pCO₂ extracted from the LDEOv2014 database (points) and generated by the artificial neural network (lines) for grid cells having more than 40 months of data. The error bars associated with the data represent the inter-annual variability, reported as the highest and lowest recorded values for a given month at a given location.

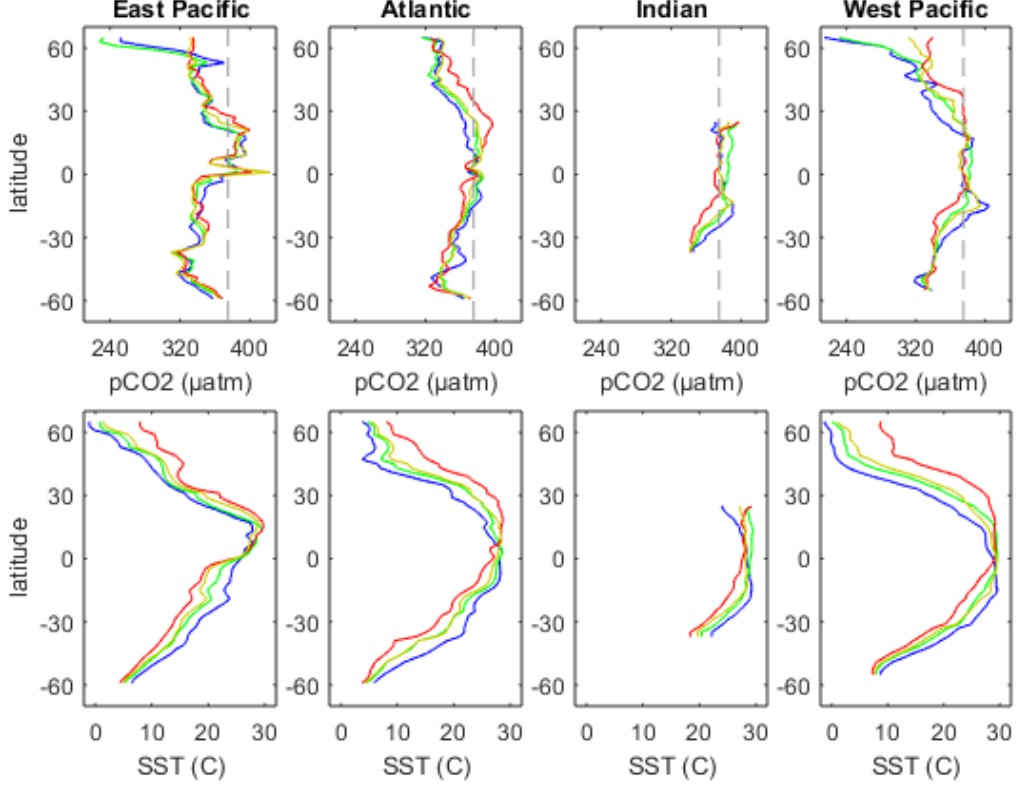

Figure 4: Seasonal-mean latitudinal profiles of pCO$_2$ (top) and SST (bottom) for the continental shelves surrounding 4 oceanic basins. Blue lines: averages over the months of January, February and March; green lines: averages over the months of April, May and June; red lines: averages over the months of July, August and September; yellow lines: averages over the months of October, November and December. The dashed line in the top panels represents the average atmospheric pCO$_2$ for year 2006.




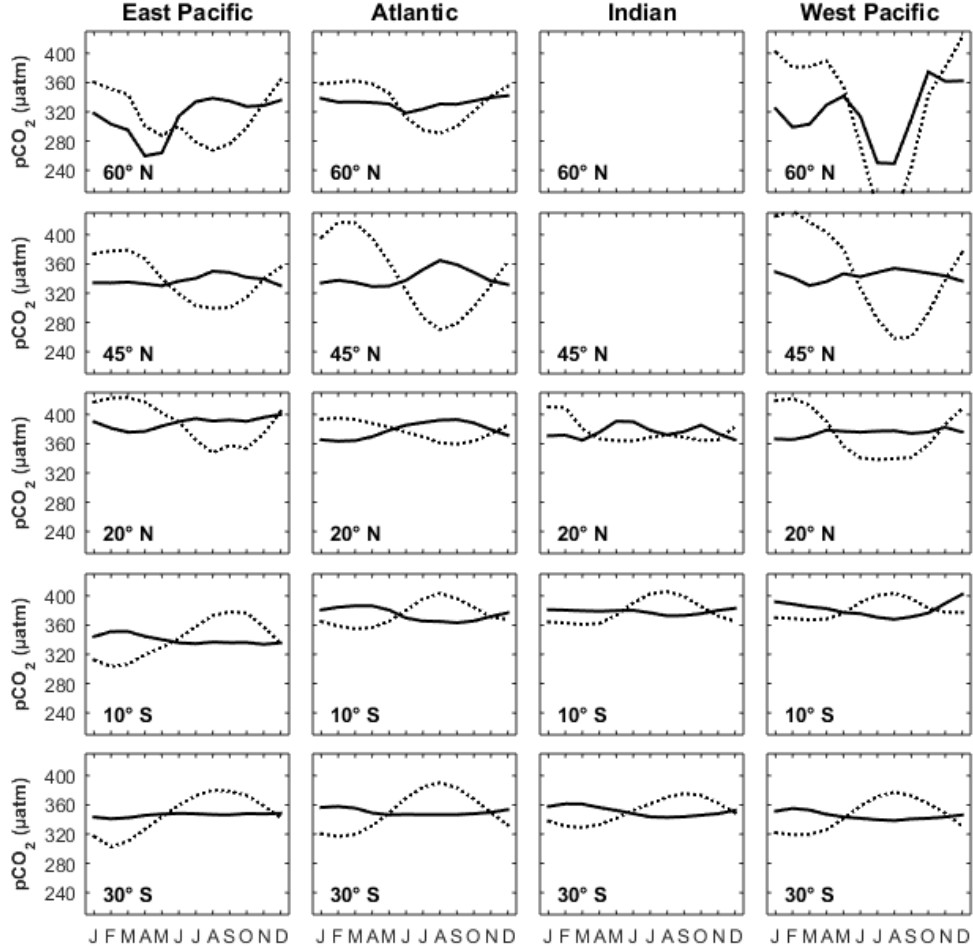

Figure 5: Seasonal cycle of observed (continuous lines) and temperature normalized $pCO_2$ (dashed lines) at 5 different latitudes in 4 oceanic basins.