# Peer review of "Global high resolution monthly pCO₂ climatology for the coastal ocean derived from neural network interpolation"

_Biogeosciences, 2017_

## Referee Comment (RC1) · R. Wanninkhof (Referee) · 4 Apr 2017

Review of bg-2017-64
 Global high-resolution monthly pCO$_2$ climatology for the coastal ocean derived from neural network interpolation by Laruelle et al.
Reviewer: Rik Wanninkhof, NOAA/AOML

This is largely a descriptive paper of procedures to create monthly estimates of coastal pCO$_2$ levels. As mentioned in the abstract, Laruelle et al. use a modified version of a two-step artificial neural network method (SOM-FFN) to interpolate the pCO$_2$ data along the continental margins with a spatial resolution of 0.25 degrees and with monthly resolution from 1998 until 2014.
The effort is clearly an impressive one and an important contribution to coastal ocean science.  However there are some shortcomings.  Many readers will not fully understand the approach and assumptions in SOM-FNN. and this needs more discussion.  The manuscript lacks in context and interpretation. Some of the procedural shortcomings that were in the initial global open ocean effort as described in Landschützer et al., (2013; 2015) prevail.

While there are comparisons and validations of the SOM-FNN approach it mostly in terms of a RMSE. It  is unclear what impact the RMSE would have on the phenomena investigated.  Other means of comparison of how well the approach works should be performed.  Rödenbeck et al (2015) present some nice diagnostics that could be applied. At very least examples of the distribution of errors in pCO$_2$ should be shown in histograms.

As the authors indicate, their definition of the coastal realm (200 nm or 1000 m depth) covers a much greater region than commonly viewed as coastal. The outer edge of the domain for much of the ocean can be considered "blue water".  Therefor it is surprising that the differences between the coastal SOM-FFNN and open ocean SOM-FNN in Landschützer et al. are large.   A more comprehensive diagnostic comparison should be made as it could suggest some fundamental issues with the approach.

The validation approach is weak.  There is significant (complete?) overlap between the data in SOCAT and that of Takahashi.  The biases in datasets are likely due to different data reduction approaches.  More comparisons should be made with actual data not used in the training, and more data should be excluded from the training for validation purposes.

It is unclear how the change in surface water over time is dealt with.  Are the pCO$_2$ data normalized like in the Takahashi monthly climatology?  SST and SSS from the WOA are used but are these monthly climatologies that do not reflect change over time.  This exercise provides monthly maps from 1998-2014 and it is clear how this is done.  Also, the product is referred to as a climatology but it sounds like it is a monthly time series.  That is, climatology mostly refers to a (multi) decadal average.

The grouping of provinces such that a coastal region can include an inshore and open ocean province is odd. Perhaps limit the coastal area to just one province
It is difficult to assess the data density for the different provinces using as validation or training.

Specific comments often relating to the general observations are below. The referenced text is in italics:

Line 125:" *motivated a number of modifications of the global ocean SOM-FFN method, including a 16 fold increase in spatial resolution from 1 degree to 0.25 degree, the introduction of a second neuron layer in the FFN calculations, the addition of new environmental variables as biogeochemical predictors, and a shortening of the simulation period to the period 1998 through 2014, rate of sea ice*
*SST, SSS, bathymetry, sea-ice concentration and chlorophyll a second artificial neuron layer*".  Some more detail on how these modification impact the results would be worthwhile .

Line 175: "*SOM-FFN from generating negative values.*"    This suggests that there are issues with the original setup.  Adding a second neuron layer to prevent negative values certainly is unorthodox.

Line 193: "*All the datasets used in our calculations were converted from their original spatial resolutions to a regular 0.25 degree resolution grid.*"  Specify what the original resolution was for each dataset.

Line 196: "*SST and SSS maps were taken from the World Ocean Atlas (Antonov et al., 2010 for SST and Locarnini et al., 2010 for SSS).*"  Are these monthly climatologies or monthly time series?  If the former it is unclear how the time element from 1998-2014 is incorporated.

Line  203 and beyond: " *validation are extracted from the LDEOv2014 database*
*The coastal SOM-FFN results are validated through a comparison with the LDEOv2014 data (Takahashi et al., 2016).*"  This is not independent data and not a proper validation in statistical sense.

Line 280: "*Considering these complexities, the achieved RMSE is quite good.*"  Two issues here.  How are the complexities determined?  That is, we know the coastal region is complex but it is unclear if the complexity is incorporated into the analysis using T, S, chl-a and sea ice. And, based on what criteria is the RMSE quite good.

Line 306:" *which compares with the most robust pCO2 regional coastal estimates from the literature (Chen et al., 2016)*". Chen et al. 2016 use a crude remote sensing approach. These are by no means "most robust".

Line 349: "*highlight the current knowledge gap regarding the mean state and variability of the transition zone.* " It is unclear if this highlights a knowledge gap or highlights issues wit the SOM_FNN approach. This warrants some discussion

Line 358: "*Our results indicate that the very nearshore processes controlling the CO2 dynamics likely*" Again the SOM-FNN is a mathematical construct. So I guess what the authors are stating is that the SOM-FNN cannot address adequately nearshore dynamics.

Line 429 "☐☐☐☐2 ". The "n" generally refers to salinity normalization. Perhaps use pCO2(SSTmean) .

Line 470: "*cells at a 0.25° spatial resolution for each of the 204 month of the simulation period (from January 1998 to December 2014). Climatologically averaged pCO2 maps for each month are*". The use of the term climatology is ambiguous here.

Line 471: The province names are peculiar "Deep Polar, Polar Very deep Polar" Table 1 suggests that Ice is a predictor in the tropics? Also P3 and P4 appear to have the same "distribution".

Figure 1 shows a peculiar extension off of New Zealand. Is this the Chatham Rise and is this considered coastal?

Figure 2: Perhaps comment on the absence of high $pCO_2$ in the SOM-FNN for the summer monsoon upwelling region in the Arabian Sea. Data of the Takahashi climatology clearly show this. Figure 2 does not show the high $pCO_2$ Arabian Sea seasonal (JAS) upwelling off the coast of the Arabian Peninsula.

---

## Referee Comment (RC2) · Anonymous Referee #2 · 14 Apr 2017

Review of bg-2017-64 "Global high-resolution monthly pCO2 climatology for the coastal ocean derived from neural network interpolation" by Laruelle et al.

This manuscript proposed a modified two-step artificial neural network method for deriving pCO2 (SOM-FFN, Landschützer et al., 2013), and focused on shelf seas. The most important modification are (1) much higher resolution as 0.25 degree; (2) inclusion of sea-ice as a predictor of pCO2. From this effort, the authors may present a fine scale coastal sea pCO2 globally, as Fig. 2 in the manuscript shown. This is certainly of value. However, there are some major issues. The method is not new, rather an interpolation of the open ocean model. It was said that all data were converted to 0.25 degree from their original resolution. Then please indicate clearly original resolution

of each data, for example, SSS, SST and depth. At least for SST and SSS from the World Ocean Atlas, I wonder if the resolution is fine in the shelf seas (sorry I do not check, my memory is 1 degree). If it is true, I do not think such an interpolation of SST and SSS would help in deriving really high resolution pCO2 (i.e. the final result might be close to a simple interpolation of modeling pCO2 of 1 degree resolution). SOCAT was used for tuning the model and LDEO was used for validation, while the two dataset was largely overlapped. This is not allowed for developing a sound and solid approach. Randomly picking data from SOCAT for calibration, and then removing those data at the same location when picking the LDEO data for validation, would not be too hard to do. The target of this manuscript is not clear. Based on the title, it looks that it is talking about a new product. As to the text, methods and validation are vague, while the authors are still eager to describe the seasonality and spatial distribution, but with no way to go into depth. And maybe because of no full confidence in the results, they frequently warned "considered with caution". I would suggest the authors focusing on method and validation, teasing each detail carefully, which would raise the merit of this study. Because one of the most important changes is to include ice, the authors need to show that by including ice, what was improved? What more was acquired/learned?

Specific comments: Abstract- Writing of the abstract needs to be improved. A very clear point should be delivered. People want to know by modifying an established algorithm, what has been acquired/improved and how good it is. Now the authors just say it is assessed using two datasets. Meridional distribution is confirmed. And then talking about seasonality produced from this dataset, which people do not know if it is true or not. If spatial and temporal variability are what the authors concerned, the title should be changed correspondingly. Line 36-39, "Overall, the seasonality in shelf pCO2 cannot solely be explained by temperature-induced changes in solubility, but are also the result of seasonal changes in circulation, mixing, and biological productivity." This should be well known by everybody. I wonder what it adds to place this sentence in the abstract. It is not clear if it is to explain the seasonality the model produced is not satisfied, or simply to explain the seasonality. One may guess that in the model

only temperature was included, so the modeling seasonality can't be explained. But in fact salinity, chlorophyll and sea-ice were all included as predictors in the model, with circulation, mixing, and biological productivity all considered in addition to temperature-induced changes in solubility.

Line 118, it is Landschützer et al. 2015? Should it be 2014?

Line 141-144, "This approach facilitates future integration with existing global ocean data products (e.g., Landschützer et al., 2016; Rödenbeck et al., 2015) and model outputs, which typically struggle to represent the shallowest parts of the ocean (Bourgeois et al., 2016)". Can you explain what the inner boundary of the global ocean data products is, where they are still confident? I do not think 500 m depth would still be too shallow to struggle. I would think that using 500 m depth as the outer boundary of shelf model would be more than enough (You used 1000 m depth as the outer boundary).

Line 152-156, chlorophyll was not included to define biogeochemical provinces using SOM?

Line 185-189, SeaWiFS extends to 2014? Please double-check. To my knowledge, it ends in 2010. By the way, normally people write it as SeaWiFS, not SeaWIFS.

Line 186, should it be "one of the environmental drivers"?

Section 2.2, it would be better if to appear before the model. Then no need to ask readers to "see below" in Line 164 and 168.

Line 198, why ice was recalculated? And what kind of recalculation?

Line 211-222 is not evaluation. It is the model training.

Line 216, do you mean you used chlorophyll in FFN but not in SOM? Why?

I would say that the entire data and method section is really confusing. A cartoon, with input and out clearly indicated, and calibration (training) and validation clearly separated, would help. Also, why twice FFN? The rationale to do this is not clear.

[Figure]

Line 353-359, this explanation is confusing. There is no reason why results from the global open ocean model can be so different from the coastal model in the overlapped cells. The only critical changes are higher resolution (actually it is an interpolation) and sea ice. Have you tried giving up ice, let other conditions be the same, see what it will be?

Fig. 2, suggest to use other color, say brown for lands. It is now not easy to tell ice cover from the land.

---

## Author Comment (AC1) · 19 Jun 2017

Review of bg-2017-64

Global high-resolution monthly pCO$_2$ climatology for the coastal ocean derived from neural network interpolation by Laruelle et al.

Reviewer: Rik Wanninkhof, NOAA/AOML

This is largely a descriptive paper of procedures to create monthly estimates of coastal pCO$_2$ levels. As mentioned in the abstract, Laruelle et al. use a modified version of a two-step artificial neural network method (SOM-FFN) to interpolate the pCO$_2$ data along the continental margins with a spatial resolution of 0.25 degrees and with monthly resolution from 1998 until 2014.

The effort is clearly an impressive one and an important contribution to coastal ocean science. However there are some shortcomings. Many readers will not fully understand the approach and assumptions in SOM-FNN. and this needs more discussion. The manuscript lacks in context and interpretation. Some of the procedural shortcomings that were in the initial global open ocean effort as described in Landschützer et al., (2013; 2015) prevail.

We are grateful for the reviewer's evaluation and his constructive suggestions. Please find bellow a detailed answer to each comment. All our answers are written in.
On behalf of all co-authors,

Goulven Laruelle

We have introduced a new section to the manuscript, which critically discusses the strength and weaknesses of the approach and its changes since the first open ocean version from Landschützer et al. (2013). This new section permits to better appraise the improvements achieved by tailoring the oceanic set-up for the coastal region and identify the remaining knowledge gaps.

We further understand that one of the main reviewer's concerns relates to the choice of validating the results using a database that largely overlaps with the one used to calibrate the model. Following his recommendation, we modified our approach and, using the latest versions of both SOCAT (i.e. version 4) and LDEO (i.e. v2015), we have now created two entirely independent datasets: one for the calibration (named SOCAT*) and one for validation (LDEO*). These two datasets were generated by randomly assigning each measurement common to both original databases to either SOCAT* or LDEO* (see comment 3 below for further details on the new approach). Another important suggestion was to further elaborate on the comparison between the simulated pCO$_2$ field and the validation dataset. We thus created new maps displaying the mean residuals errors between the pCO$_2$ values generated by the SOM_FFN, on the one hand, and those extracted from LDEO* and SOCAT*, on the other hand. This representation allows for a more detailed analysis of the performance of the model. As suggested by the reviewer, histograms of residual errors were also computed for each biogeochemical province and will be discussed in the updated manuscript. In addition, we have also introduced a new predictor (wind speed), which helped improve the performances of the SOM_FFN compared to those presented in the previous version of the manuscript.

While there are comparisons and validations of the SOM-FNN approach it mostly in terms of a RMSE. It is unclear what impact the RMSE would have on the phenomena investigated. Other means of comparison of how well the approach works should be performed. Rödenbeck et al (2015) present some nice diagnostics that could be applied.

At very least examples of the distribution of errors in pCO₂ should be shown in histograms.

[1] We agree with the reviewer that the assessment of the performance of the model only relied on averaged biases and RMSE calculated for each biogeochemical province. In the updated manuscript, we propose to include maps presenting the average residual errors between the pCO₂ field generated by the model and pCO₂ data extracted from the calibration (SOCAT*) and validation (LDEO*) datasets. They are obtained by subtracting the observed values from model output in each grid cell for every month where observations are available. This representation not only allows to assess which regions provide the best match with the observations but also to identify where the simulated pCO₂ overestimates (positive values, in red on the figure below) or underestimates (negative values, in blue on the figure below) the field data. Moreover, as suggested by the reviewer, we introduce a new figure, presenting the distribution of the residual errors between the results of the SOM_FFN and LDEO* for each biogeochemical province. This figure reveals nearly Gaussian distributions of the residuals for every biogeochemical province with the exception of province P8, for which the spread is not only the highest (indicating the largest discrepancy between model and observations), but also slightly skewed toward high values, thus revealing a tendency to overestimate the observed pCO₂.

[Figure]

Figure 1: Mean residuals calculated as the difference between the SOM_FFM pCO₂ outputs and pCO₂ observations from SOCAT* (top) and LDEO* (bottom).

[Figure]

Figure: Histograms reporting the distribution of residuals between observed (LDEO*) and computed (SOM_FFN) pCO₂ in each biogeochemical province.

As the authors indicate, their definition of the coastal realm (200 nm or 1000 m depth) covers a much greater region than commonly viewed as coastal. The outer edge of the domain for much of the ocean can be considered "blue water". Therefor it is surprising that the differences between the coastal SOM-FFNN and open ocean SOM-FNN in Landschützer et al. are large. A more comprehensive diagnostic comparison should be made as it could suggest some fundamental issues with the approach.

[2] Although both the coastal SOM_FFN presented in this study and the oceanic SOM_FFN published in Landschützer et al. have a significant overlapping domains, they were not trained with the same datasets. For the most part, the coastal data from SOCAT used here to calibrate our model were not included in the data pool used for the open ocean simulations. In addition, the characteristic ranges of values within which both models are trained are also different for some of the environmental parameters. In particular, the average bathymetry and sea surface salinities are often significantly lower for data used. It is thus not surprising to observe significant differences between the results produced by both models, yet we agree with the reviewer that the magnitude of difference is somewhat interesting and highlights current knowledge gaps regarding the coastal ocean to open ocean transition zone. This certainly deserves some further investigation; however, we do believe that this is beyond the scope of this study. Nevertheless, in the updated manuscript, we will further discuss the differences between coastal and open SOM-FFN in the transition zone.

The validation approach is weak. There is significant (complete?) overlap between the data in SOCAT and that of Takahashi. The biases in datasets are likely due to different data reduction approaches. More comparisons should be made with actual data not used in the training, and more data should be excluded from the training for validation purposes.

[3] As mentioned by the reviewer, the SOCAT and LDEO databases have a large overlap, and the two datasets cannot be considered independent. In order to provide robust calibration and validation we now created two fully independent datasets based on SOCAT and LDEO, which do not contain any common measurement. We used the latest releases of both databases (i.e. SOCATv4 and LDEOv2015) and filtered out all non-coastal data points, as was already done in the previous version of the manuscript. Under our definition of the coastal zone, SOCATv4 contains $\sim 8 \ 10^6$ data points and LDEO $\sim 5.6 \ 10^6$, over 70% of which are also part of SOCATv4. We then randomly assigned each of those common data point to either database to insure that each data only belongs to one dataset. In the updated manuscript, the new datasets are renamed SOCAT* which is used to train the SOM_FFN, and LDEO* which is only used for validation purposes. In the new manuscript, the procedure used to create SOCAT* and LDEO* will be detailed in section 2.2 (Data Sources and processing).

The use of a more robust validation did not alter significantly the performances of the SOM_FFN and, combined with the inclusion of wind speed as a new predictor, the biases and RMSE generated by the model when compared with LDEO* are actually slightly lower than those presented in the original simulations (see table below). Also, note that the use of SOCATv4 and LDEOv2015 provides a significant number of data for the year 2015, which motivated us to expend our simulation period from 17 to 18 years.

[Figure]

Figure: Number of observations contained in each 0.25° grid cell of the SOCAT* (top) and LDEO* (bottom) databases.

Table: Root mean squared error between observed and calculated $pCO_2$ in the different biogeochemical provinces. The SOM-FFN results are compared to data extracted from the SOCAT* and the LDEO* databases.

| Province | SOCAT* Bias (µatm) | SOCAT* RMSE (µatm) | LDEO* Bias (µatm) | LDEO* RMSE (µatm) |
|---|---|---|---|---|
| P1 | 0.0 | 19.1 | 2.0 | 20.5 |
| P2 | 0.2 | 24.7 | 1.3 | 27.2 |
| P3 | -0.3 | 16.1 | 2.3 | 22.7 |
| P4 | -0.2 | 31.2 | -1.6 | 33.0 |
| P5 | 0.0 | 34.2 | -1.4 | 38.0 |
| P6 | 0.0 | 24.3 | 1.3 | 27.9 |
| P7 | 0.1 | 37.2 | -0.2 | 52.5 |
| P8 | 0.2 | 46.8 | 3.9 | 51.4 |
| P9 | -0.1 | 23.0 | -2.5 | 33.4 |
| P10 | 0.0 | 35.7 | 1.6 | 53.1 |
| Global | 0.0 | 32.9 | 0.0 | 39.2 |

It is unclear how the change in surface water over time is dealt with. Are the $pCO_2$ data normalized like in the Takahashi monthly climatology? SST and SSS from the WOA are used but are these monthly climatologies that do not reflect change over time. This exercise provides monthly maps from 1998-2014 and it is clear how this is done. Also, the product is referred to as a climatology but it sounds like it is a monthly time series. That is, climatology mostly refers to a (multi) decadal average.

[4] During the training of the SOM_FFN, all $pCO_2$ data from SOCAT* are associated to a set of environmental conditions corresponding to the location and moment in time when the $pCO_2$ was measured. The relationships linking $pCO_2$ to environmental conditions as established by the FFN are then applied in each cell of the simulation domain for each of the 216 month of the simulation period. The inputs used for these calculations are 3 dimensional fields (latitude, longitude and time) containing values for each grid cell at every monthly time step. We will make sure to clarify this procedure in the updated manuscript. All the data used as inputs for both SOM and FFN are thus monthly times series and no normalization was applied to the data as was performed in Takahashi et al. (2009).

We realize that our frequent use of the word climatology may be misleading as to what our product really is. In the updated manuscript and the abstract, we will state more clearly that our calculations are performed for every month of the simulation period and thus produce monthly maps for each of the years simulated. Only then, a monthly climatology is derived from those results.

Also note that, in the new simulations, SST and SSS data are not taken from the World Ocean Atlas anymore but from the Met Office's EN4: quality controlled subsurface ocean temperature and salinity profiles and objective analyses (Good et al., 2009). This change was implemented following a comment from reviewer #2 regarding mismatches in spatial resolution of some datasets (the new SST/SSS datasets are at the spatial resolution of 0.25 degree as opposed to WOA which only provides values at 1 degree).

The grouping of provinces such that a coastal region can include an inshore and open ocean province is odd. Perhaps limit the coastal area to just one province

[5] The biogeochemical provinces generated by the Self Organizing Maps regroup ensembles of cells together because of similarities in their environmental characteristics. Within each biogeochemical province, however, some variability can be found and, while bathymetry may significantly contribute to the grouping of cells within a given province, so do the other environmental parameters (i.e. SSS, SST, wind speed and sea ice). As a consequence, some provinces have an extension that includes nearshore and more open waters but for which the range of temperature for example might be limited (see figure below displaying the spatial extent of the updated biogeochemical provinces). The choice to use the SOM and divide the coastal ocean into several provinces as was done for the open ocean in Landschützer et al. (2013) was motivated by the large variety of environmental settings that can be found in the coastal ocean. The current number of 10 provinces was selected as the optimal number during the calibration phase. When developing the model, several simulations were performed with the SOM using increasing numbers of biogeochemical provinces (from 6 to 20) and 10 was the number of biogeochemical provinces yielding the best results in terms of RMSE when compared with both SOCAT and LDEO databases. This number of biogeochemical provinces also guarantees that sufficient data will be located in each biogeochemical province, thus insuring both a proper training of the algorithm and the possibility of a validation against a significant number of observations. For instance, the spatio-temporal distribution of the biogeochemical provinces used in our last simulation allows for at least 1000 different grid cells to be used for validation against LDEO*.

[Figure]

**Biogeochemical provinces**

Figure 1: Map of the 10 different biogeochemical provinces generated by the SOM.

It is difficult to assess the data density for the different provinces using as validation or training.

[6] We understand the reviewer's concern and agree that, in the original version of the manuscript, limited information was provided regarding the spatial distribution of the

pCO$_2$ data used for calibration or validation. In the updated manuscript, a new figure (see comment [3]) now shows the data density of the SOCAT* and LDEO* databases for each grid cell of the simulation domain, thus providing a clear view of the amount and spatial distribution of data used both for calibration and validation..

Specific comments often relating to the general observations are below. The referenced text is in italics:
Line 125:" *motivated a number of modifications of the global ocean SOM-FFN method, including a 16 fold increase in spatial resolution from 1 degree to 0.25 degree, the introduction of a second neuron layer in the FFN calculations, the addition of new environmental variables as biogeochemical predictors, and a shortening of the simulation period to the period 1998 through 2014, rate of sea ice SST, SSS, bathymetry, sea-ice concentration and chlorophyll a second artificial neuron layer*". Some more detail on how these modification impact the results would be worthwhile .

[7] As mentioned by both reviewers, the different modifications introduced compared to the original set-up of the global ocean SOM_FFN are only mentioned in our method section but not discussed in details in our results. In the updated manuscript, we discuss the impact of those modifications (i.e. resolution and new predictors such as sea ice and wind speed). For instance, the added value of performing our simulations at the spatial resolution of 0.25° is discussed using examples such as the ability of our model to capture the plumes of larges rivers such as the Amazon, which produces an area located North of its river mouth characterized by pCO$_2$ values significantly lower than those of the surrounding waters (Cooley et al., 2007; Ibanez et al., 2015). The new discussion will also involve the addition of results from simulations performed only using SST, SSS, bathymetry and chlorophyll as predictors (as suggested by reviewer #2). The results of those simulations are presented in the table below and allow quantifying how the addition of new predictors affects the performance of the model. For instance, it can be noticed that, overall, the global RMSE increase significantly (from 39.2 to 48 µatm in the comparison with LDEO* when chlorophyll, sea ice and wind speed are not taken into account and from 39.2 to 45 µatm when only sea ice and wind speed are not taken into account). This deterioration of the performance of the model, however, is not evenly affecting all provinces and it can be observed in particular that provinces located at high latitudes (i.e. P8, P9 and P10) perform significantly worse without the inclusion of wind speed and sea ice.

Table: Biases and root mean squared error (RMSE) between observed and calculated pCO$_2$ using only SST, SSS and bathymetry (STB) or SST, SSS, bathymetry and chlorophyll (STBC) as predictors.

| Province | SOCAT* | | | | LDEO* | | | |
| | Bias (µatm) | | RMSE (µatm) | | Bias (µatm) | | RMSE (µatm) | |
| | STB | STBC | STB | STBC | STB | STBC | STB | STBC |
|---|---|---|---|---|---|---|---|---|
| P1 | 0.0 | -0.2 | 20.8 | 21.0 | 2.4 | 2.0 | 21.7 | 21.5 |
| P2 | -0.1 | 0.1 | 26.9 | 27.8 | 0.5 | 0.8 | 29.0 | 29.6 |
| P3 | 0.0 | -0.5 | 22.7 | 21.3 | 3.0 | 2.3 | 27.1 | 26.8 |
| P4 | 0.0 | -0.2 | 33.0 | 33.0 | -1.7 | -2.3 | 33.8 | 33.8 |
| P5 | 0.2 | 0.1 | 52.7 | 42.2 | -1.7 | -0.9 | 56.9 | 44.5 |
| P6 | 0.0 | 0.1 | 26.8 | 26.5 | -0.5 | 0.6 | 28.9 | 28.0 |
| P7 | 0.4 | 0.3 | 44.3 | 44.1 | 1.2 | 0.3 | 59.3 | 58.8 |

| | | | | | | | |
|---|---|---|---|---|---|---|---|
| **P8** | 0.1 | 0.4 | 82.6 | 80.0 | 9.1 | 9.0 | 56.3 | 58.5 |
| **P9** | 0.1 | 0.9 | 34.7 | 36.5 | -2.6 | -2.8 | 39.8 | 41.8 |
| **P10** | -0.3 | 0.7 | 49.8 | 49.5 | -3.9 | -3.0 | 76.5 | 75.4 |
| **Global** | 0.1 | 0.2 | 43.9 | 42.4 | 0.0 | 0.0 | 48.0 | 45.0 |

Line 175: "*SOM-FFN from generating negative values.*" This suggests that there are issues with the original setup. Adding a second neuron layer to prevent negative values certainly is unorthodox.

[8] The SOM-FFN method is some form of a non-linear regression model, which in cases of bad conditioning also produces out of range values. We point here that the negative values do not suggest issues with the model as a whole, but rather issues with the setup of the model. While we did not face the problem of negative values using a standard hidden layer in the open ocean, the added complexity combined with little data in certain province can cause this behaviour in coastal seas. For instance, there exist very few measurements for shallows waters with very low salinity and high sea ice coverage. Faced with conditions for which it was not trained, the SOM_FFN does not perform ideally and may generate unrealistic values. In our original manuscript we solved this by introducing a second hidden layer of neurons, however, we found a more stable solution in terms of negative values, i.e we replaced the second neuron layer with the use of a sigmoid activation function bounded between 0 and 1 (normalized $pCO_2$ units) in the hidden layer. This means that per definition our results are bound to stay above 0. The implementation of this solution did not deteriorate the overall results but prevented the SOM_FFN from generating negative $pCO_2$ values. The new simulations for the revised manuscript were thus carried out with this new setting, which now only uses a single neuron layer.

Line 193: "*All the datasets used in our calculations were converted from their original spatial resolutions to a regular 0.25 degree resolution grid.*" Specify what the original resolution was for each dataset.

[9] A more thorough description of the datasets used in the study will be included into section 2.2 (Data Sources and processing). This description explicitly states the original temporal and spatial resolution of each dataset used. This information will be compiled in the new table reported below. In addition, for the sake of reproducibility, a link toward all datasets used will be provided in the 'Data Availability' section at the end of the manuscript. Note that, as already reported in comment 4, all our products have now an original resolution of 0.25° or finer.

Table: Datasets used to create the environmental forcing files. The original spatial and temporal resolution and the main manipulations applied for their use in the SOM_FFN are also reported.

| **Predictor** | **dataset** | **resolution** | **reference** | **Manipulation** |
|---|---|---|---|---|
| SST | EN4 | 0.25°, daily | Good et al., 2013 | Monthly average |
| SSS | EN4 | 0.25°, daily | Good et al., 2013 | Monthly average |
| Bathymetry | ETOPO2 | 2 minutes | US Department of Commerce, | Aggregation to 0.25° |

| | | | 2006 | |
|---|---|---|---|---|
| Sea ice | NSIDC | 0.25°, monthly | Cavalieri et al., 1996 | Monthly rate of change in sea ice coverage |
| Chlorophyll a | SeaWifs, MODIS | 9km, monthly | NASA, 2016 | Aggregation to 0.25° |
| Wind speed | ERA | 0.25°, 6hours | Dee et al., 2011 | Monthly average |

Line 196: "*SST and SSS maps were taken from the World Ocean Atlas (Antonov et al., 2010 for SST and Locarnini et al., 2010 for SSS)*." Are these monthly climatologies or monthly time series? If the former it is unclear how the time element from 1998-2014 is incorporated.

[10] The new simulations do not use SST and SSS from the World Ocean Atlas anymore but from the Met Office's EN4: quality controlled subsurface ocean temperature and salinity profiles and objective analyses (Good et al., 2009). Those data are time series and contain individual values in each grid cell of the simulation domain for each of the 216 month of the simulation period. Additional information regarding the incorporation of the time element in our calculation is included in answer [4] and the updated manuscript will be more explicit with respect to the way our calculations are performed.

Line 203 and beyond: " *validation are extracted from the LDEOv2014 database The coastal SOM-FFN results are validated through a comparison with the LDEOv2014 data (Takahashi et al., 2016)*." This is not independent data and not a proper validation in statistical sense.

[11] As discussed in the answer to reviewer's comment [3], we fully agree that the original validation was significantly weakened by the large overlap between SOCAT and LDEO. Now that we created two entirely independent datasets to train the model (SOCAT*) and evaluate its performances (LDEO*), we believe that the term "validation" is now appropriate for the updated manuscript.

Line 280: "*Considering these complexities, the achieved RMSE is quite good*." Two issues here. How are the complexities determined? That is, we know the coastal region is complex but it is unclear if the complexity is incorporated into the analysis using T, S, chl-a and sea ice. And, based on what criteria is the RMSE quite good.

[12] It is true that the coastal region is known to be a complex environment and that was the main message of this sentence. Whether our analysis capture the intricate complexity of the coastal zone has to be indeed better discussed in the revised manuscript. We will thus further develop the section dedicated to the discussion and quantification of the effects induced by modifications in SOM_FFN configuration on its performance (see answer to comment [7]). With respect to the RMSE, our criteria to consider the performance of our model 'quite good' is the comparison with the RMSE reported in regional studies . This is further discussed in the answer [13] below.

Line 306:" *which compares with the most robust pCO2 regional coastal estimates from the literature (Chen et al., 2016*)". Chen et al. 2016 use a crude remote sensing approach. These are by no means "most robust".

[13] The paper by Chen et al. (2016) indeed presents $pCO_2$ fields for the Western Florida shelf generated using remote sensing. Such methodology certainly is different and arguably less sophisticated that the method described in our study. However, we did not mean to directly compare the performance of our model with those of Chen et al. (2016). Our aim was to find as many recent studies as possible to compare our results and to gain some confidence in our estimates. Their study reports (table 1, page 12) a list of regional coastal models generating $pCO_2$ fields derived from other environmental factors. Although the methods used in this list varies greatly (including Mutiple Linear Regressions, Mechanistic semi analytical models and Self Organizing Maps), we believe it was relevant to confront the performance of our model applied globally with those of other coastal models, which are only applied at regional scale in well covered areas. What we meant to say is that there exist a body of literature using various methodological approaches to generate $pCO_2$ fields and the article by Chen was mostly used for his table. Nevertheless, following the reviewer's comment, we will tone down our statement that our results compare with the most robust estimates from the literature. Rather, we'll state that the RMSE calculated in our best constrained biogeochemical provinces (i.e. in the 20-30 µatm range for P1, P2, P3 and P6) can be compared with those obtained by regional models applied in well monitored areas.

Line 349: "*highlight the current knowledge gap regarding the mean state and variability of the transition zone.* " It is unclear if this highlights a knowledge gap or highlights issues wit the SOM_FNN approach. This warrants some discussion

[14] We agree with the reviewer's comment (as well as similar concerns' raised by reviewer 2) and recognise that the original version of the manuscript only briefly compared the results of the updated coastal SOM_FFN with those of the original oceanic model. In the updated manuscript, a more in depth comparison with the results of the open ocean configuration will be provided. This will allow better identifying the added value of the modifications done to the SOM_FFN method in our study and help clearly identify remaining knowledge gaps.

Line 358: "*Our results indicate that the very nearshore processes controlling the CO2 dynamics likely*" Again the SOM-FNN is a mathematical construct. So I guess what the authors are stating is that the SOM-FNN cannot address adequately nearshore dynamics.

[15] The reviewer is correct; this sentence was meant to stress that, in spite of the improvement provided by the new method, some very nearshore processes still cannot be addressed perfectly. As the reviewer pointed out, the problem does not lie with the mathematical approach used by the spatial resolution required to capture very nearshore processes. The sentence was rephrased as follows:
"***Overall, the occurrence of large residuals in the shallowest cells of our calculation domain in*** our results ***(fig. 2) suggest*** that the very nearshore processes controlling the $CO_2$ dynamics likely are the most difficult to reproduce ***at the global scale.***"

Line 429 "2 ". The "n" generally refers to salinity normalization. Perhaps use pCO2(SSTmean) .

[16] We will follow the reviewer's suggestion in the updated manuscript and use $pCO_2(SSTmean)$ instead of $npCO_2$.

Line 470: "*cells at a 0.25° spatial resolution for each of the 204 month of the simulation period (from January 1998 to December 2014). Climatologically averaged pCO2 maps for each month are*". The use of the term climatology is ambiguous here.

[17] We agree with the reviewer, the term climatology is ambiguous in this sentence and elsewhere. To avoid any confusion, the paragraph was rephrased as follows:

"The data product associated to this manuscript consists of a netcdf file containing the $pCO_2$ for ice-free cells at a 0.25° spatial resolution for each of the **_216_** month of the simulation period (from January 1998 to December **_2015_**). **_12 maps representing mean pCO₂ fields calculated for each month over the simulation period_** are also provided."

Line 471: The province names are peculiar "Deep Polar, Polar Very deep Polar"

[18] Our choice of names for the different biogeochemical provinces was only meant to outline their main geographical distribution. Both reviewers commented on the lack of added value of the distributions of the biogeochemical provinces. In the updated manuscript, the biogeochemical provinces will only be referred to as P1, P2 and so on to avoid confusion. Section 3.1 however, will still discuss the spatial extent of the each biogeochemical province.

Table 1 suggests that Ice is a predictor in the tropics?

[19] We agree that the use of Ice as predictor in the tropics is not relevant, however Ice cover in the tropics in our predictor dataset was 0 at all times, and hence it did not influence the neural network. To avoid confusion, in the updated simulation, Ice is only a predictor in provinces P5 to P10, in which at least partial seasonal ice coverage is reported.

Table 2: List of the biogeochemical provinces, their geographic distribution and the environmental predictors used to calculate surface ocean $pCO_2$. SSS stands for sea surface salinity, SST for sea surface temperature, Bathy for bathymetry, Ice for sea-ice cover, Chl for chlorophyll concentration **_and Wind for wind speed_**.

| Province | SSS | SST | Bathy | Ice | Chl | Wind |
|----------|-----|-----|-------|-----|-----|------|
| P1 | X | X | X |   | X | X |
| P2 | X | X | X |   | X | X |
| P3 | X | X | X |   | X | X |
| P4 | X | X | X |   | X | X |
| P5 | X | X | X | X | X | X |
| P6 | X | X | X | X | X | X |
| P7 | X | X | X | X | X | X |

| P8 | X | X | X | X | X |
| P9 | X | X | X | X | X |
| P10 | X | X | X | X | X |

Also P3 and P4 appear to have the same "distribution".

[20] In the original simulations, provinces P3 and P4 did not display exactly the same spatio-temporal distribution but were both referred to as "Deep Tropical" which could indeed lead to confusion. Actually, the average water depth of cells included in P4 was deeper than that of those included in P3 and, P4 generally characterized more 'open waters'. As mentioned in answer [5], the updated manuscript will describe and discuss the spatial distributions of the 10 biogeochemical provinces but the restrictive 'distributions' will be removed from table 1.

Figure 1 shows a peculiar extension off of New Zealand. Is this the Chatham Rise and is this considered coastal?

[21] The extension Southward and Eastward of New Zealand are the Campbell Plateau and Chatham Rise, respectively. They are considered coastal following our 'extended' definition of the continental shelf and upper slope because they are characterized by depth shallower than 1000m (our outer limit) and connected to a continental platform.

Figure 2: Perhaps comment on the absence of high $pCO_2$ in the SOM-FNN for the summer monsoon upwelling region in the Arabian Sea. Data of the Takahashi climatology clearly show this. Figure 2 does not show the high $pCO_2$ Arabian Sea seasonal (JAS) upwelling off the coast of the Arabian Peninsula.

[22] It is true that high $pCO_2$ values have been regularly observed along the coast of the Arabian Sea (Sarma et al., 2003) and are considered to be the consequence of monsoon driver upwelling occurring in the region. As noted by the reviewer, the SOM-FFN does not reproduce these oversaturated waters. We now mention and discuss the inability of the SOM_FFN to reproduce this known feature of the Arabian shelf in section 3.3.1, which discusses the general spatial patterns of the $pCO_2$ fields generated by the model.

**Literature cited in the responses:**

Cavalieri, D. J., Parkinson, C. L., Gloersen, P., and Zwally, H.: Sea Ice Concentrations from Nimbus-7 SMMR and DMSP SSM/I-SSMIS Passive Microwave Data, years 1990–2011, NASA DAAC at the Natl. Snow and Ice Data Cent., Boulder, Colo. (Updated yearly.), 1996.

Chen, S., Hu, C., Byrne, R. H., Robbins, L. L., and Yang, B.: Remote estimation of surface pCO2 on the West Florida Shelf, Continental Shelf Research, 128, 10–25, 2016.

Cooley, S. R., V. J. Coles, A. Subramaniam, and P. L. Yager (2007), Seasonal variations in the Amazon plume-related atmospheric carbon sink, Global Biogeochem. Cycles, 21, GB3014, doi:10.1029/2006GB002831.

Dee, D. P., Uppala, S. M., Simmons, A. J., Berrisford, P., Poli, P., Kobayashi, S., Andrae, U., Balmaseda, M. A., Balsamo, G., Bauer, P., Bechtold, P., Beljaars, A. C. M., van de Berg, L., Bidlot, J., Bormann, N., Delsol, C., Dragani, R., Fuentes, M., Geer, A. J., Haimberger, L., Healy, S. B., Hersbach, H., Hòlm, E. V., Isaksen, L., Kallberg, P., Köhler, M., Matricardi, M., Mcnally, A. P., Monge-Sanz, B. M., Morcrette, J. J., Park, B. K., Peubey, C., de Rosnay, P., Tavolato, C., Thépaut, J. N. and Vitart, F.: The ERA-Interim reanalysis: Configuration and performance of the data assimilation system, Q. J. R. Meteorol. Soc., 137(656), 553–597, doi:10.1002/qj.828, 2011.

Good, S. A., M. J. Martin and N. A. Rayner, 2013. EN4: quality controlled ocean temperature and salinity profiles and monthly objective analyses with uncertainty estimates, Journal of Geophysical Research: Oceans, 118, 6704-6716, doi:10.1002/2013JC009067

Ibánhez, J. S. P., D. Diverrès, M. Araujo, and N. Lefèvre (2015), Seasonal and interannual variability of sea-air $CO_2$ fluxes in the tropical Atlantic affected by the Amazon River plume, Global Biogeochem. Cycles, 29, 1640–1655, doi:10.1002/2015GB005110.

Landschützer, P., Gruber, N., Bakker, D. C. E., Schuster, U., Nakaoka, S., Payne, M. R., Sasse, T., and Zeng, J.: A neural network-based estimate of the seasonal to inter-annual variability of the Atlantic Ocean carbon sink, Biogeosciences, 10, 7793-7815, doi:10.5194/bg-10-7793-2013, 2013.

NASA Goddard Space Flight Center, Ocean Ecology Laboratory, Ocean Biology Processing Group; (Dataset Release 2016): MODIS-Aqua chlorophyll Data; NASA Goddard Space Flight Center, Ocean Ecology Laboratory, Ocean Biology Processing Group, 2016.

Sarma, V. V. S. S., Monthly variability in surface $pCO_2$ and net air-sea $CO_2$ flux in the Arabian Sea, J. Geophys. Res., 108 (C8), 3255, doi:10.1029/2001JC001062, 2003.

Takahashi, T., S. C. Sutherland, R. Wanninkhof, C. Sweeney, R. A. Feely, D. W. Chipman, B. Hales, G. Friederich, F. Chavez, A. Watson, D. C. E. Bakker, U. Schuster, N. Metzl, H. Yoshikawa-Inoue, M. Ishii, T. Midorikawa, Y. Nojiri, C. Sabine, J. Olafsson, Th. S. Arnarson, B. Tilbrook, T. Johannessen, A. Olsen, Richard Bellerby, A. Körtzinger, T. Steinhoff, M. Hoppema, H. J. W. de Baar, C. S. Wong, Bruno Delille and N. R. Bates (2009). Climatological mean and decadal changes in surface ocean $pCO_2$, and net sea-air $CO_2$ flux over the global oceans. Deep-Sea Res. II, 56, 554-577

U.S. Department of Commerce, National Oceanic and Atmospheric Administration, National Geophysical Data Center. 2006. 2-minute Gridded Global Relief Data (ETOPO2v2). http://www.ngdc.noaa.gov/mgg/fliers/06mgg01.html. Accessed 21 May 2017.

---

## Author Comment (AC2) · 19 Jun 2017

Review of bg-2017-64 "Global high-resolution monthly pCO2 climatology for the coastal ocean derived from neural network interpolation" by Laruelle et al.

This manuscript proposed a modified two-step artificial neural network method for deriving pCO2 (SOM-FFN, Landschützer et al., 2013), and focused on shelf seas. The most important modification are (1) much higher resolution as 0.25 degree; (2) inclusion of sea-ice as a predictor of pCO2. From this effort, the authors may present a fine scale coastal sea pCO2 globally, as Fig. 2 in the manuscript shown. This is certainly of value. However, there are some major issues. The method is not new, rather an interpolation of the open ocean model.

We are pleased to see that the reviewer values our coastal $pCO_2$ maps and are grateful for his constructive remarks and suggestions. We understand that the reviewer is not fully convinced by the novelty of the method and the added value our manuscript under its current form. As explained in answers, we do not concur with the statement that our model only is an extension or interpolation of the previously existing oceanic model. Instead, we believe that it is a significantly modified version, specifically tailored to reconstruct the complex coastal $pCO_2$ cycle. In the updated manuscript, we propose to put more emphasis on the modifications of the original SOM_FFN and compare our coastal set up with the open ocean one. Further attention will also be given to better quantifying the improvements resulting from the modification of the open ocean set-up from Landschützer et al. (2013) and identifying the remaining knowledge gaps (see also replies to comments 2, 7, 14 of reviewer 1).

The reviewer was also concerned by the weakness of the validation of our results performed using a database that largely overlaps with the database used to calibrate the model. Following both reviewer's recommendations, we modified our approach and, using the latest versions of both SOCAT (i.e. version 4) and LDEO (i.e. v2015), we created two entirely independent datasets, named SOCAT* for the calibration and LDEO* for the validation. These two datasets were generated by randomly assigning each measurement common to both original databases to either SOCAT* or LDEO* (see comment 2 below for further details on the new approach). In addition, we have also introduced a new predictor (wind speed), which helped improve the performances of the SOM_FFN compared to those presented in the previous version of the manuscript.

Please find bellow a detailed answer to each comment. All our answers are written in blue and the modifications within the text are highlighted in bold and italic.

On behalf of all co-authors,

Goulven Laruelle

It was said that all data were converted to 0.25 degree from their original resolution. Then please indicate clearly original resolution of each data, for example, SSS, SST and depth. At least for SST and SSS from the World Ocean Atlas, I wonder if the resolution is fine in the shelf seas (sorry I do not check, my memory is 1 degree). If it is true, I do not think such an interpolation of SST and SSS would help in deriving really high resolution pCO2 (i.e. the final result might be close to a simple interpolation of modeling pCO2 of 1 degree resolution).

[1] The spatial resolution of SST and SSS from the World Ocean Atlas is indeed only 1 degree. In response to the reviewer's comment, we now apply 0.25° resolution datasets

for SSS and SST by using Met Office's EN4: quality controlled subsurface ocean temperature and salinity profiles and objective analyses (Good et al., 2009). By doing so, all predictors used for the calculation of the SOM_FFN have now resolution of 0.25° or higher. We also propose the inclusion of the table below, which lists the selected datasets used, their purpose (i.e. calibration, validation…) and original spatio-temporal resolution.

We reiterate here that we disagree with the notion that our model is a mere interpolation of the global oceanic model developed by Landschützer et al. (2013). Although both the coastal SOM_FFN presented in this study and the oceanic SOM_FFN published in Landschützer et al. (2013) share common methodologies, they were not trained with the same datasets. For the most part, the coastal data from SOCAT used here for calibration and validation was not included in the data pool used for the open ocean simulations. In addition, the ranges of values (within which both models are trained) are also different for some of the environmental parameters. In particular, the average bathymetry and sea surface salinities are often significantly lower in coastal regions than in the open ocean. We thus believe that the important physical and biogeochemical differences between coastal and open oceanic waters fully justify careful retraining of the SOM_FFN. In addition, the typical spatial scales of physical and biogeochemical gradients in nearshore waters are often smaller than 1 degree and justify the implementation of the SOM_FFN at a higher resolution. Nevertheless, to better demonstrate the value of our approach, we follow the comment of the reviewer and discuss in more details the comparison between open and coastal ocean models in the revised manuscript.

Table 1: Datasets used to create the environmental forcing files. The original spatial and temporal resolution and the main manipulations applied for their use in the SOM_FFN are also reported.

| Predictor | dataset | resolution | reference | Manipulation |
|---|---|---|---|---|
| SST | EN4 | 0.25°, daily | Good et al., 2013 | Monthly average |
| SSS | EN4 | 0.25°, daily | Good et al., 2013 | Monthly average |
| Bathymetry | ETOPO2 | 2 minutes | US Department of Commerce, 2006 | Aggregation to 0.25° |
| Sea ice | NSIDC | 0.25°, daily | Cavalieri et al., 1996 | Monthly rate of change in sea ice coverage |
| Chlorophyll a | SeaWifs, MODIS | 9km, monthly | NASA, 2016 | Aggregation to 0.25° |
| Wind speed | ERA | 0.25°, 6hours | Dee et al., 2011 | Monthly average |

SOCAT was used for tuning the model and LDEO was used for validation, while the two dataset was largely overlapped. This is not allowed for developing a sound and solid approach.Randomly picking data from SOCAT for calibration, and then removing those data at the same location when picking the LDEO data for validation, would not be too hard to do.

[2] As mentioned by the reviewer, the SOCAT and LDEO databases have a large overlap, and the two datasets cannot be considered independent. In order to remedy to this problem, we followed the reviewer suggestion and created two datasets based on SOCAT and LDEO which do not contain any common measurements. We used the latest releases of both databases (i.e. SOCATv4 and LDEOv2015) and filtered out all non-coastal data points, as it was already done in the previous version of the manuscript. Under our definition of the coastal zone, SOCATv4 contains ~8 $10^6$ data points and LDEO ~5.6 $10^6$, over 70% of which are also part of SOCATv4. We then randomly assigned each of those common data points to either database, thus insuring that each data only belongs to one dataset. In the updated manuscript, the new datasets are then called SOCAT* which is used to train the SOM_FFN, and LDEO* which is only used for validation purposes. In the new manuscript, the procedure used to create SOCAT* and LDEO* will be detailed in section 2.2 (Data Sources and processing).

The use of a more robust validation did not alter significantly the performances of the SOM_FFN and, combined with the inclusion of wind speed as a new predictor, the biases and RMSE generated by the model when compared with LDEO* are actually slightly lower than those presented in the original simulations (see table below). Also, note that the use of SOCATv4 and LDEOv2015 provides a significant number of data for the year 2015, which motivated us to expend our simulation period from 17 year to 18.

[Figure]

Figure: Number of observations contained in each 0.25° grid cell of the SOCAT* (top) and LDEO* (bottom) databases.

"Table: Root mean squared error between observed and calculated $pCO_2$ in the different biogeochemical provinces. The SOM-FFN results are compared to data extracted from the SOCAT* and the LDEO* databases.

| Province | SOCAT* | | LDEO* | |
|---|---|---|---|---|
| | Bias (µatm) | RMSE (µatm) | Bias (µatm) | RMSE (µatm) |
| P1 | 0.0 | 19.1 | 2.0 | 20.5 |
| P2 | 0.2 | 24.7 | 1.3 | 27.2 |
| P3 | -0.3 | 16.1 | 2.3 | 22.7 |
| P4 | -0.2 | 31.2 | -1.6 | 33.0 |
| P5 | 0.0 | 34.2 | -1.4 | 38.0 |
| P6 | 0.0 | 24.3 | 1.3 | 27.9 |
| P7 | 0.1 | 37.2 | -0.2 | 52.5 |
| P8 | 0.2 | 46.8 | 3.9 | 51.4 |
| P9 | -0.1 | 23.0 | -2.5 | 33.4 |
| P10 | 0.0 | 35.7 | 1.6 | 53.1 |
| Global | 0.0 | 32.9 | 0.0 | 39.2 |

The target of this manuscript is not clear. Based on the title, it looks that it is talking about a new product. As to the text, methods and validation are vague, while the authors are still eager to describe the seasonality and spatial distribution, but with no way to go into depth. And maybe because of no full confidence in the results, they frequently warned "considered with caution". I would suggest the authors focusing on method and validation, teasing each detail carefully, which would raise the merit of this study. Because one of the most important changes is to include ice, the authors need to show that by including ice, what was improved? What more was acquired/learned?

[3] The manuscript presents monthly $pCO_2$ fields for the coastal ocean generated by a statistical method that was never applied in such environment. Obviously, a large part of the manuscript is dedicated to presenting the methods (i.e. the modifications of the open ocean set up in order to better capture the dynamics of continental shelves) and we agree with the reviewer that each critical point of the method should be discussed thoroughly. Following his recommendation, we now discuss results obtained with our model ignoring our new predictors (wind speed and sea ice cover) to better quantify their contribution to the accuracy of our results. Similarly, the added value of performing our simulations at the spatial resolution of 0.25° is also discussed using examples such as the ability of our model to capture the plumes of larges rivers such as the Amazon, which produces an area located North of its river mouth characterized by $pCO_2$ values significantly lower than those of the surrounding waters (Cooley et al., 2007; Ibanez et al., 2015). We believe that this discussion will clearly allow the reader to understand the added value of our approach. In addition, the validation of our results is now much more developed by including maps of mean residuals obtained when comparing the $pCO_2$ field generated by the SOM_FFN with data from LDEO* and histograms of the distribution of these residuals with each biogeochemical province (see figures below).

However, we also believe that it is useful to thoroughly describe our results in terms of spatial and seasonal trends and not restrict our analysis to comparison against

validation data. One of the main values of our data product is the resolution of the seasonal variations of $pCO_2$ in regions of the continental shelf that were largely under sampled until now. We thus believe that, although the main purpose of our manuscript is to describe a new coastal $pCO_2$ data product, dedicating a significant fraction of our results and discussion to the emerging spatial and temporal patterns in the coastal $pCO_2$ field is justified and relevant. As for our warning that results in certain regions should be "considered with caution": Despite the increasing number of observations collected and the methodological advancements, there are still regions, such as the Siberian shelves, where only few observations exist and our process understanding is limited. Limited observations mean on the one hand limited information to train our model but on the other hand also only limited means to validate our results. This should not be misinterpreted as us having a lack of confidence, but rather us having limited means of validating our results for some areas of the global coastal ocean. With this statement, we wanted to highlight these limitations and help the reader to critically reflect on our results.

Table: Biases and root mean squared error (RMSE) between observed and calculated $pCO_2$ using only SST, SSS and bathymetry (STB) or SST, SSS, bathymetry and chlorophyll (STBC) as predictors.

| Province | SOCAT* | | | | LDEO* | | | |
| --- | --- | --- | --- | --- | --- | --- | --- | --- |
| | Bias (µatm) | | RMSE (µatm) | | Bias (µatm) | | RMSE (µatm) | |
| | STB | STBC | STB | STBC | STB | STBC | STB | STBC |
| P1 | 0.0 | -0.2 | 20.8 | 21.0 | 2.4 | 2.0 | 21.7 | 21.5 |
| P2 | -0.1 | 0.1 | 26.9 | 27.8 | 0.5 | 0.8 | 29.0 | 29.6 |
| P3 | 0.0 | -0.5 | 22.7 | 21.3 | 3.0 | 2.3 | 27.1 | 26.8 |
| P4 | 0.0 | -0.2 | 33.0 | 33.0 | -1.7 | -2.3 | 33.8 | 33.8 |
| P5 | 0.2 | 0.1 | 52.7 | 42.2 | -1.7 | -0.9 | 56.9 | 44.5 |
| P6 | 0.0 | 0.1 | 26.8 | 26.5 | -0.5 | 0.6 | 28.9 | 28.0 |
| P7 | 0.4 | 0.3 | 44.3 | 44.1 | 1.2 | 0.3 | 59.3 | 58.8 |
| P8 | 0.1 | 0.4 | 82.6 | 80.0 | 9.1 | 9.0 | 56.3 | 58.5 |
| P9 | 0.1 | 0.9 | 34.7 | 36.5 | -2.6 | -2.8 | 39.8 | 41.8 |
| P10 | -0.3 | 0.7 | 49.8 | 49.5 | -3.9 | -3.0 | 76.5 | 75.4 |
| Global | 0.1 | 0.2 | 43.9 | 42.4 | 0.0 | 0.0 | 48.0 | 45.0 |

[Figure]

Figure 1: Mean residuals calculated as the difference between the SOM_FFM pCO$_2$ outputs and pCO$_2$ observations from SOCAT* (top) and LDEO* (bottom).

[Figure]

Figure: Histograms reporting the distribution of residuals between observed (LDEO*) and computed (SOM_FFN) pCO$_2$ in each biogeochemical province.

Specific comments: Abstract- Writing of the abstract needs to be improved. A very clear point should be delivered. People want to know by modifying an established algorithm, what has been acquired/improved and how good it is. Now the authors just say it is assessed using two datasets. Meridional distribution is confirmed. And then talking about seasonality produced from this dataset, which people do not know if it is true or not. If spatial and temporal variability are what the authors concerned, the title should be changed correspondingly.

[4] As mentioned in answer [3], the updated manuscript now dedicates more effort to better identifying what was improved and learned with each of the modification introduced to the SOM_FFN compared to its open ocean set up. Also, we now implemented a more robust validation of our results (following several suggestions of both reviewers), including a revised comparison with monthly climatological cycles extracted from LDEO* at 40 locations (see figure below). We thus do not agree with the reviewer when he suggests that our discussion regarding the seasonality of $pCO_2$ in coastal waters is unsubstantiated. We not only think that these seasonal signals are supported by our validation but also that the discussion of the seasonal dynamics of the coastal $pCO_2$ is very relevant to the manuscript and the wider research community. We agree however that the original abstract was not specific enough (especially with respect to seasonal variability) and we will make sure that the updated abstract better reflects the novelty of our approach.

[Figure]

Figure: Climatological monthly mean pCO₂ extracted from the LDEO* database (points) and generated by the artificial neural network (lines) for grid cells having more than 40 months of data. The error bars associated with the data represent the inter-annual variability, reported as the highest and lowest recorded values for a given month at a given location.

Line 36-39, "Overall, the seasonality in shelf pCO2 cannot solely be explained by temperature-induced changes in solubility, but are also the result of seasonal changes in circulation, mixing, and biological productivity."

This should be well known by everybody. I wonder what it adds to place this sentence in the abstract. It is not clear if it is to explain the seasonality the model produced is not satisfied, or simply to explain the seasonality. One may guess that in the model only temperature was included, so the modeling seasonality can't be explained. But in fact salinity, chlorophyll and sea-ice were all included as predictors in the model, with circulation, mixing, and biological productivity all considered in addition to temperatureinduced changes in solubility.

[5] We agree with the reviewer that all readers familiar with the dynamics of carbon in coastal waters will be aware that the seasonal changes of $pCO_2$ are not only driven by temperature variations but also hydrodynamics, planktonic productivity etc... The purpose of this sentence was to refer to our analysis of the effect of temperature change on the seasonal cycle of $pCO_2$ presented at the end of section 3 but we agree that the phrasing was too generic and did not report any new finding. In the updated manuscript, the abstract will be more specific and the outcome of our seasonal analysis more clearly presented (i.e. in which regions of the world, is temperature the dominant driver of the seasonal change in coastal $pCO_2$, see also answer [4]).

Line 118, it is Landschützer et al. 2015? Should it be 2014?
[6] Indeed, the SOM FFN method is only briefly described in Landschützer et al. (2015) and the reference will be replaced by Landschützer et al. (2014) in the updated manuscript.

Line 141-144, "This approach facilitates future integration with existing global ocean data products (e.g., Landschützer et al., 2016; Rödenbeck et al., 2015) and model outputs, which typically struggle to represent the shallowest parts of the ocean (Bourgeois et al., 2016)". Can you explain what the inner boundary of the global ocean data products is, where they are still confident? I do not think 500 m depth would still be too shallow to struggle. I would think that using 500 m depth as the outer boundary of shelf model would be more than enough (You used 1000 m depth as the outer boundary).
[7] Unfortunately, there is no such thing as a universally accepted inner boundary for ocean data products and models but the extension of their simulation domain varies from one study to the other. The 200m isobaths if commonly used as limit between the open ocean and continental shelves but this limit is somewhat artificial (Walsh et al., 1988, Laruelle et al., 2013). The purpose of extending our outer limit for coastal water as far as 1000m depth is to insure an overlap between coastal and oceanic data product to prevent some regions of the world to remain untreated by either approaches.

Line 152-156, chlorophyll was not included to define biogeochemical provinces using SOM?
[8] Indeed, chlorophyll was not included to define the biogeochemical provinces using SOM, due to the fact that the data coverage is incomplete in the high latitudes in winter due to e.g. cloud coverage. This is the same reason Chl a is excluded from the calculations of provinces P8, P9 and P10 during the Feed Forward Network step. This will be clarified in the text.

Line 185-189, SeaWiFS extends to 2014? Please double-check. To my knowledge, it ends in 2010. By the way, normally people write it as SeaWiFS, not SeaWIFS.

[9] As pointed out by the reviewer, SeaWiFS data do not extend past 2010. The data used later than this date and all the way to December 2015 are taken for MODIS. Also, SeaWiFS will be written as suggested by the reviewer throughout the updated manuscript. This reply will be used to clarify the manuscript and details will be included in the table listing all data sources (see answer [1]).

Line 186, should it be "one of the environmental drivers"?
[10] The sentence will be corrected as suggested.

Section 2.2, it would be better if to appear before the model. Then no need to ask readers to "see below" in Line 164 and 168.
[11] Following the reviewer's suggestion, the sections 2.1 and 2.2 of the manuscript will be inverted, in order to present the datasets used and their processing, before describing the modifications performed to the SOM_FFN.

Line 198, why ice was recalculated? And what kind of recalculation?
[12] The original spatial resolution of the sea ice coverage is days and monthly averages had to be calculated from the original data as well as monthly rates of change in sea ice coverage.  This is now explained more clearly in the updated manuscript. In addition, the new table 2 listing all the original spatial and temporal resolutions of all datasets and the manipulations performed with them will help make the data processing more transparent.

Line 211-222 is not evaluation. It is the model training.
[13] Following the reviewer's suggestion, this subsection has been renamed 'model training'.

Line 216, do you mean you used chlorophyll in FFN but not in SOM? Why?
[14] Indeed chlorophyll was used in FFN but not in SOM as justified in answer [8].

I would say that the entire data and method section is really confusing. A cartoon, with input and out clearly indicated, and calibration (training) and validation clearly separated, would help. Also, why twice FFN? The rationale to do this is not clear.

[15] We agree with the reviewer that the different steps and datasets required by our approach may be confusing to the reader and we now improved the clarity of the method in the updated manuscript. In particular, the suggestion of the reviewer to use a conceptual scheme detailing the different steps of the method will be included in the revised ms.
As for the choice of using twice the FFN, it is true that such choice is uncommon and generally not required in a Feedforward Network. Following the remarks of both reviewers regarding this modification, another solution was considered to replace the second neuron layer with the use of a sigmoid activation function bounded between 0 and 1 in the hidden layer. The implementation of this solution did not deteriorate the overall results. The new simulations were thus carried out with this new setting which only uses a single neuron layer.

Line 353-359, this explanation is confusing. There is no reason why results from the global open ocean model can be so different from the coastal model in the overlapped

cells. The only critical changes are higher resolution (actually it is an interpolation) and sea ice. Have you tried giving up ice, let other conditions be the same, see what it will be?

[16] As mentioned in answer [1], we do not agree with the notion that our results are just an interpolation of the oceanic model. Other than the spatial resolution and the choice of environmental predictors, both oceanic and coastal models were trained on fundamentally different datasets – the open ocean model was trained with open ocean $pCO_2$ measurements and the coastal model was trained with coastal $pCO_2$ measurements. Therefore, we are not surprised that the 2 estimates differ in overlapping areas. However, we do agree that the magnitude of disagreement is somewhat larger than one would expect, highlighting on the one hand current knowledge gaps regarding the coastal to open ocean continuum and on the other hand that more research is needed to close this knowledge gap. The suggestion from the reviewer to perform simulations without the new coastal predictors to quantify their effect is now also included in the updated manuscript, as already discussed in answer [3].

Fig. 2, suggest to use other color, say brown for lands. It is now not easy to tell ice cover from the land.

[17] The suggestion of the reviewer has been implemented in the new version of the manuscript. As an example, all the maps presented in these replies already use a brown colour to represent land.

**Literature cited in the responses:**

Cavalieri, D. J., Parkinson, C. L., Gloersen, P., and Zwally, H.: Sea Ice Concentrations from Nimbus-7 SMMR and DMSP SSM/I-SSMIS Passive Microwave Data, years 1990–2011, NASA DAAC at the Natl. Snow and Ice Data Cent., Boulder, Colo. (Updated yearly.), 1996.

Cooley, S. R., V. J. Coles, A. Subramaniam, and P. L. Yager (2007), Seasonal variations in the Amazon plume-related atmospheric carbon sink, Global Biogeochem. Cycles, 21, GB3014, doi:10.1029/2006GB002831.

Dee, D. P., Uppala, S. M., Simmons, A. J., Berrisford, P., Poli, P., Kobayashi, S., Andrae, U., Balmaseda, M. A., Balsamo, G., Bauer, P., Bechtold, P., Beljaars, A. C. M., van de Berg, L., Bidlot, J., Bormann, N., Delsol, C., Dragani, R., Fuentes, M., Geer, A. J., Haimberger, L., Healy, S. B., Hersbach, H., Hòlm, E. V., Isaksen, L., Kallberg, P., Köhler, M., Matricardi, M., Mcnally, A. P., Monge-Sanz, B. M., Morcrette, J. J., Park, B. K., Peubey, C., de Rosnay, P., Tavolato, C., Thépaut, J. N. and Vitart, F.: The ERA-Interim reanalysis: Configuration and performance of the data assimilation system, Q. J. R. Meteorol. Soc., 137(656), 553–597, doi:10.1002/qj.828, 2011.

Good, S. A., M. J. Martin and N. A. Rayner, 2013. EN4: quality controlled ocean temperature and salinity profiles and monthly objective analyses with uncertainty estimates, Journal of Geophysical Research: Oceans, 118, 6704-6716, doi:10.1002/2013JC009067

Ibánhez, J. S. P., D. Diverrès, M. Araujo, and N. Lefèvre (2015), Seasonal and interannual variability of sea-air CO2 fluxes in the tropical Atlantic affected by the Amazon River plume, Global Biogeochem. Cycles, 29, 1640–1655, doi:10.1002/2015GB005110.

Landschützer, P., Gruber, N., Bakker, D. C. E., Schuster, U., Nakaoka, S., Payne, M. R., Sasse, T., and Zeng, J.: A neural network-based estimate of the seasonal to inter-annual variability of the Atlantic Ocean carbon sink, Biogeosciences, 10, 7793-7815, doi:10.5194/bg-10-7793-2013, 2013.

Landschützer, P., Gruber, N., Bakker, D. C. E., and Schuster, U.: Recent variability of the global ocean carbon sink, Global Biogeochemical Cycles, 28, 927–949, doi:10.1002/2014GB004853, 2014.

Landschützer, P., Gruber, N., Haumann, F. A. Rödenbeck, C. Bakker, D.C.E. , van Heuven, S. Hoppema, M., Metzl, N., Sweeney, C., Takahashi, T., Tilbrook, B. and Wanninkhof, R.: The reinvigoration of the Southern Ocean carbon sink, Science, 349, 1221-1224. doi: 10.1126/science.aab2620, 2015.

Laruelle, G. G., Dürr, H. H., Lauerwald, R., Hartmann, J., Slomp, C. P., Goossens, N., and Regnier, P. A. G.: Global multi-scale segmentation of continental and coastal waters from the watersheds to the continental margins, Hydrol. Earth Syst. Sci., 17, 2029-2051, doi:10.5194/hess-17-2029-2013, 2013.

NASA Goddard Space Flight Center, Ocean Ecology Laboratory, Ocean Biology Processing Group; (Dataset Release 2016): MODIS-Aqua chlorophyll Data; NASA Goddard Space Flight Center, Ocean Ecology Laboratory, Ocean Biology Processing Group, 2016.

U.S. Department of Commerce, National Oceanic and Atmospheric Administration, National Geophysical Data Center. 2006. 2-minute Gridded Global Relief Data (ETOPO2v2). http://www.ngdc.noaa.gov/mgg/fliers/06mgg01.html. Accessed 21 May 2017.

Walsh, J. J.: On the nature of continental shelves, Academic Press, San Diego, New York, Berkeley, Boston, London, Sydney, Tokyo, Toronto, 1988.